# Tonotopic and non-auditory organization of the mouse dorsal inferior colliculus revealed by two-photon imaging

Aaron Benson Wong, J Gerard G Borst*

Department of Neuroscience, Erasmus MC, University Medical Center Rotterdam, Rotterdam, Netherlands

**Abstract** The dorsal (DCIC) and lateral cortices (LCIC) of the inferior colliculus are major targets of the auditory and non-auditory cortical areas, suggesting a role in complex multimodal information processing. However, relatively little is known about their functional organization. We utilized *in vivo* two-photon $Ca^{2+}$ imaging in awake mice expressing GCaMP6s in GABAergic or non-GABAergic neurons in the IC to investigate their spatial organization. We found different classes of temporal responses, which we confirmed with simultaneous juxtacellular electrophysiology. Both GABAergic and non-GABAergic neurons showed spatial microheterogeneity in their temporal responses. In contrast, a robust, double rostromedial-caudolateral gradient of frequency tuning was conserved between the two groups, and even among the subclasses. This, together with the existence of a subset of neurons sensitive to spontaneous movements, provides functional evidence for redefining the border between DCIC and LCIC.

DOI: https://doi.org/10.7554/eLife.49091.001

*For correspondence:
g.borst@erasmusmc.nl

Competing interests: The authors declare that no competing interests exist.

## Introduction

The inferior colliculus (IC) is a major auditory processing center, which receives input from most brainstem auditory nuclei. The IC is usually divided into three main regions: the central nucleus (CNIC), the lateral cortex (LCIC), and the dorsal cortex (DCIC; *Oliver, 2005*). The latter two, together also known as the 'shell region' of the IC, are part of the non-lemniscal pathway, and are heavily innervated by feedback projections from the cerebral cortex. The DCIC is often defined as the part that covers the CNIC dorsally (e.g. *Zhou and Shore, 2006*). The LCIC is characterized by distinct neurochemical modules, which can be visualized with various histochemical methods such as acetylcholinesterase or NADPH diaphorase staining, as well as dense labeling of GABAergic terminals and sparse calretinin immunoreactivity (*Lesicko et al., 2016*; *Dillingham et al., 2017*). These modules show a distinct complement of Eph-ephrin guidance molecules during development (*Wallace et al., 2016*; *Gay et al., 2018*) and receive input from non-auditory areas (*Lesicko et al., 2016*; *Patel et al., 2017*), hinting at their integral role in multisensory integration in the IC. Here, we refer to the external cortex and lateral nucleus of the IC as the lateral cortex (LCIC; *Loftus et al., 2008*), which includes layer three or the ventrolateral nucleus, and refer to the dorsal cortex and pericentral nucleus as the dorsal cortex of the IC (DCIC). The DCIC and LCIC are at the surface of the mouse brain, making the dorsal IC an accessible structure for *in vivo* imaging (*Geis and Borst, 2013*; *Barnstedt et al., 2015*; *Babola et al., 2018*).

Because of the prominence of descending, cortical input, the DCIC and LCIC is ideally studied in awake, behaving animals. A number of studies have addressed the firing of IC neurons in awake bats (e.g. *Xie et al., 2005*; *Xie et al., 2007*; *Xie et al., 2008*; *Andoni and Pollak, 2011*; *Gittelman and Li, 2011a*; *Gittelman and Pollak, 2011b*; *Gittelman et al., 2012*) or mice (*Gittelman et al., 2013*; *Grimsley et al., 2013*; *Duque and Malmierca, 2015*; *Ayala et al., 2016*; *Grimsley et al., 2016*;

*Longenecker and Galazyuk, 2016*; *Galazyuk et al., 2017*). However, the yield of such recordings is limited, and the acute nature of some of the studies means that residual anesthetics and analgesics may have interfered with neuronal activity.

Moreover, relatively little is known about different cell types in the DCIC and LCIC. Around one-fourth of neurons in the IC are GABAergic (*Schofield and Beebe, 2019*). In particular the large GABAergic neurons seem to form a distinct subclass (*Ito et al., 2009*; *Ito and Oliver, 2012*; *Geis and Borst, 2013*), but based on histology, at least four different subclasses of GABAergic neurons can be discriminated, which all contribute to the ascending projections to the medial geniculate body of the thalamus (*Beebe et al., 2018*).

Here, we describe the use of two transgenic mouse lines to characterize GABAergic and glutamatergic neuronal subpopulations in the dorsal IC in awake animals using *in vivo* two-photon Ca$^{2+}$ imaging. We studied GABAergic neurons with a Gad2-IRES-Cre mouse (*Taniguchi et al., 2011*) that was crossed with the GCaMP6s reporter line Ai96 (*Madisen et al., 2015*) and a sub-population of non-GABAergic neurons using the Thy1-driven GCaMP6s transgenic line GP4.3 (*Dana et al., 2014*). We show a rich diversity of sound-evoked responses in both GABAergic and non-GABAergic neurons in the awake mouse, confirmed with simultaneous juxtacellular electrophysiology in awake animals. Remarkably, we observed a reversal of the characteristic frequency (CF) gradient in the rostromedial-caudolateral direction, which was conserved between GABAergic and non-GABAergic cells, as well as among cells with different classes of sound-evoked response. Moreover, we found a subset of neurons that were responsive to spontaneous movement of the animal, and were potentially associated with multisensory neurochemical modules (*Lesicko et al., 2016*). These findings suggest that the large majority of the dorsal IC surface belongs to the LCIC.

## Results

### Expression of GCaMP6s in IC subpopulations of transgenic mice

To better understand the response heterogeneity observed in IC neurons, we used transgenic lines in which the GCaMP reporter was selectively expressed in subpopulations. Two transgenic mouse lines were used: GP4.3, a Thy1-driven GCaMP6s expression line (*Dana et al., 2014*), and a cross between Gad2-IRES-Cre line (*Taniguchi et al., 2011*) and Ai96, a Cre-dependent GCaMP6s reporter line (*Madisen et al., 2015*). We will refer to the latter as Gad2;Ai96.

To confirm the neurochemical identity of GCaMP6s-positive neurons in the transgenic lines, we performed immunolabeling of GAD67. *Figure 1* shows example staining and the proportion of GCaMP-positive neurons expressing GAD67 in each line. We did not distinguish between the different subregions of the IC in this analysis. The large majority (4651/4898; 95%) of GCaMP-positive neurons in the GP4.3 line were not positive for GAD67 (*Figure 1A–D*), showing that the Thy1-promoter preferentially expressed GCaMP6s in non-GABAergic cells in the IC. Not all non-GABAergic cells were GCaMP-positive, as shown by the NeuN-positive, GCaMP-negative cell bodies in *Figure 1C* (arrowheads). In particular, we observed that GCaMP6s-positive neurons are enriched at the border of neurochemical modules characterized by dense GAD67 terminals (*Figure 1E–G*).

In contrast, we detected GAD67 immunoreactivity in >93% (1702/1828) of GCaMP-positive neurons in the Gad2;Ai96 mice (*Figure 1D,H–O*). Not all GAD67-positive cells in this mouse line were labeled by GCaMP6s (e.g. arrowheads in *Figure 1J*).

In addition to GAD67, the calcium buffer parvalbumin (PV) is another marker for neurochemical modules in the rat (*Chernock et al., 2004*), while the calcium buffer calretinin (CR) shows a complementary expression (i.e. in the extramodular region) in early postnatal and juvenile mice (*Dillingham et al., 2017*). To get a better idea of the relative contribution of modular and extramodular regions to GCaMP6 responses, we investigated the co-expression of GCaMP6 with PV or CR in the two transgenic lines (*Figure 1—figure supplement 1*). We found that a large majority (94%) of GCaMP+ cells in GP4.3 did not show strong immunoreactivity for either marker (CR++: 2.2%; PV++: 2.9%; PV++CR++: 0.9%), whereas in the Gad2;Ai96 line a larger fraction of GCaMP+ cells were positive for CR or PV (CR++: 17%; PV++: 22%; PV++CR++: 7%; *Figure 1—figure supplement 1G*). On the other hand, GCaMP was expressed in 38% of CR++ cells and 70% of PV++ cells in the IC of Gad2;Ai96 mice, while a lower fraction of CR++ (9.1%) and PV++ (13.4%) cells were positive for GCaMP in GP4.3 mice (*Figure 1—figure supplement 1H,I*). At the single neuron level, the presence

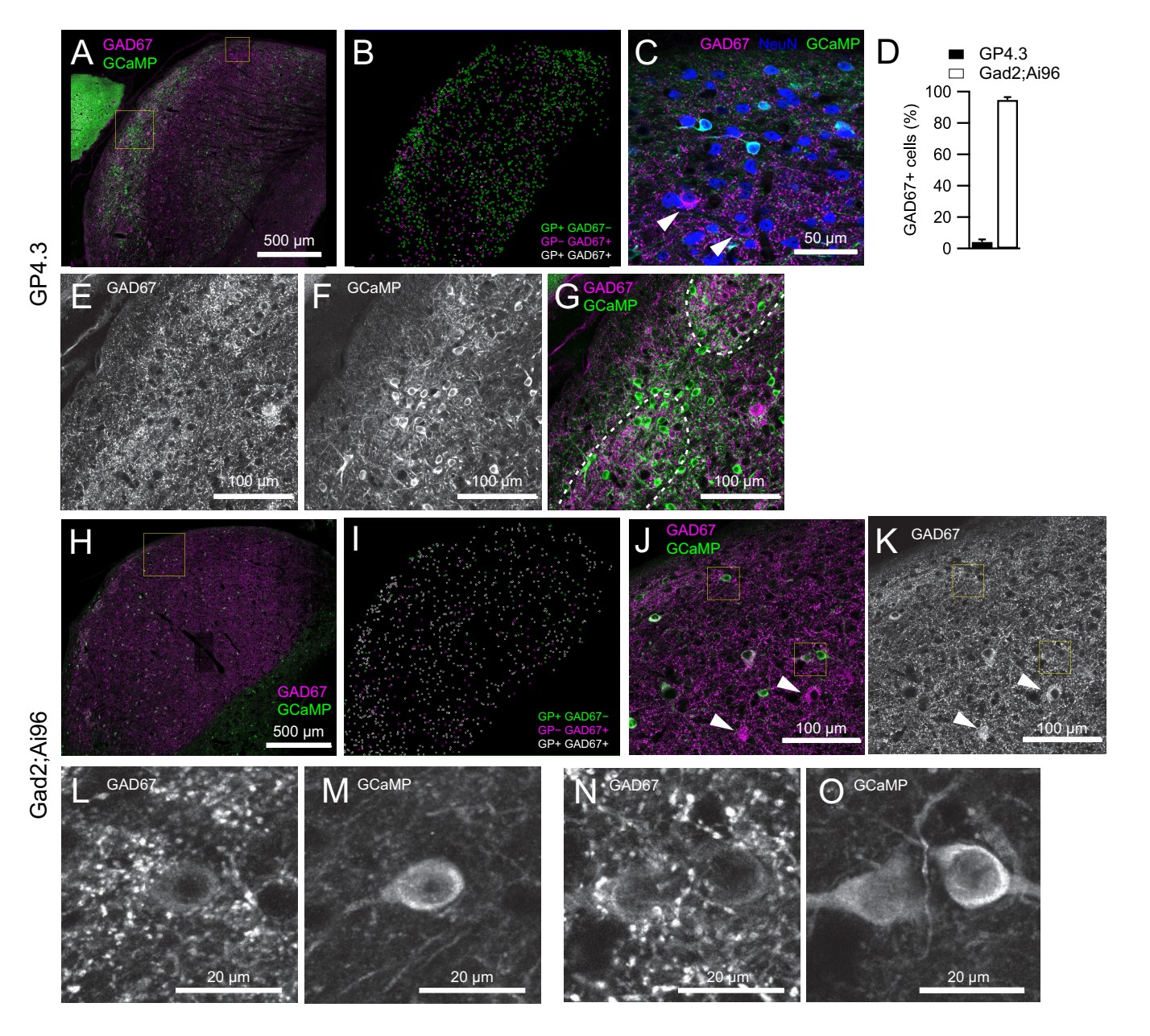

**Figure 1.** Co-expression between GCaMP6s and GAD67 in the two transgenic mouse lines. (**A**) Single optical section of an IC brain slice from a GP4.3 mouse immunolabeled for GAD67 (magenta) and GCaMP6s (green). (**B**) Distribution of GCaMP+ and GAD67+ cells within the 40 μm brain slice in **A**, color-coded by their immunoreactivity to GFP and GAD67 antibodies. GP+/−: GCaMP-positive/negative. (**C**) Enlarged image from A (small square), showing different combinations of immunoreactivity: GCaMP (green), GAD67 (magenta) and NeuN (blue). Arrowheads point at two GP−GAD67+ cells. Due to optical sectioning only a subset of cells marked in **B** are visible. (**D**) Percentage of GCaMP+ cells showing immunoreactivity for GAD67 antibodies (error bars: s.e.m.; GP4.3: n = 3 slices from three animals; Gad2;Ai96: n = 4 slices from three animals) (**E–G**) Enlarged image from large square in A showing two GAD67-dense modules marked by dashed lines. GP+ cells in GP4.3 were concentrated at the border of these modules. (**H–I**) Same as A and B, but from a Gad2;Ai96 mouse. (**J–K**) Enlarged region from H (square). Arrowheads point at two GP−GAD67+ cells. (**L–O**) Examples of GP+GAD67+ cells from G showing images from GAD67 and GCaMP channels separately.

DOI: https://doi.org/10.7554/eLife.49091.002

The following source data and figure supplement are available for figure 1:

**Source data 1.** CellCounter source data XML files generated by Cell counter plug-in in FIJI containing the actual counted datapoints of each cell type, with text files explaining the different cell types being counted.

DOI: https://doi.org/10.7554/eLife.49091.004

*Figure 1 continued on next page*

*Figure 1 continued*

**Figure supplement 1.** Expression of parvalbumin and calretinin in GCaMP+ neurons in the two transgenic mouse lines.
DOI: https://doi.org/10.7554/eLife.49091.003

or absence of the two buffers was insufficient to indicate whether the neurons lay within a module. These results suggest that for both lines there is not a clear enrichment of GCaMP6+ neurons within the neurochemical modules.

## Sound-evoked changes in $F_{GCaMP}$ of IC neurons show prominent inhibitory responses

Unanaesthetized, head-fixed mice with implanted cranial windows (*Figure 2A–B*) were imaged with two photon microscopy (*Figure 2C–D*) and subjected to 1 s pure tones of various frequencies and intensities (*Figure 2E*). Average fluorescence of regions-of-interest (ROIs) encompassing the cell body were analyzed, after subtracting the fluorescent change of a background region surrounding the cell body (*Figure 2D*). The use of longer stimuli allowed a detailed characterization of the kinetics of the fluorescence change ($\Delta F_{GCaMP}$). Importantly, we observed both increases, which were either transient or sustained, and decreases in $F_{GCaMP}$ during the presentation of stimuli, and some cells showed a sharp increase in $F_{GCaMP}$ directly after the cessation of the sound (*Figure 2E*). The different fluorescence responses were classified as excitation, inhibitory and offset, respectively (see Materials and methods, *Figure 3A–D*). The excitatory responses were further divided into onset and sustained classes based on the kinetics of the fluorescence change (*Figure 3A,B*, *Figure 3—figure supplement 1*). The decrease in fluorescence upon sound stimulation was likely due to sound-evoked inhibition of spontaneous firing in these cells. Some cells showed a mixture of response kinetics, some even to the same stimulus. Particularly common was the combination of inhibitory and offset responses (e.g. *Figure 3E,M*). Another common combination were onset-offset cells in which a lower frequency elicited an onset/sustained response, while a slightly higher frequency elicited an offset response (e.g. *Figure 3I*). Inhibitory and offset responses were not reported in earlier imaging experiments performed in anesthetized mice (*Ito et al., 2014*; *Barnstedt et al., 2015*); their use of shorter stimuli did not allow a distinction between onset and sustained responses. A great variety of frequency response areas (FRAs) was observed (*Figure 3F–M*), with some cells showing responses to tones as high as 64 kHz, the highest frequency presented in this study (e.g. *Figure 3G*).

In GP4.3 mice, 64% of the cells (696 out of 1090) cells showed detectable fluorescence changes in response to 1 s pure tone stimuli. We classified the responses and the overall FRA of a cell based on the criteria described in the Materials and methods. Among the responsive cells, 30% (210 cells) showed excitatory FRAs (e.g. *Figure 3—figure supplement 2*), with fluorescence increasing during sound presentation; while 13% (89 cells) showed offset FRAs, with fluorescence increasing after sound presentation. Another 30% (207 cells) showed inhibitory FRAs, with fluorescence decreasing during sound. The FRAs of the remaining 27% (190 cells) showed mixtures of these response categories. The proportion of different FRA classes are summarized in *Figure 4A*. The proportion of cells showing sound-evoked inhibition was clearly higher than the 2–23% reported for anesthetized C57BL/6 mice in *Willott et al. (1988a)*.

In Gad2;Ai96 mice 73% (247 out of 342 cells) of GCaMP6s-positive IC neurons showed consistent fluorescence responses to 1 s pure tone stimuli. Among the responsive cells, 46% (115 cells) showed excitatory FRAs. Four percent (11 cells) showed offset FRAs. Another 26% (65 cells) showed inhibitory FRAs. The remaining 23% (56 cells) showed mixtures of the three response categories. The proportion of different FRA classes in Gad2;Ai96 mice are summarized in *Figure 4B*; it was similar to that of GP4.3 mice, but with a somewhat smaller fraction of offset and a larger fraction of excitatory FRAs.

## Relationship between cell type and soma size

GABAergic cells had a larger soma size, as reported previously (cat: *Oliver et al., 1994*; rat: *Merchán et al., 2005*). The estimated diameters for Gad2;Ai96 and GP4.3 cells, defined as $4/\pi$ times the square root of the ROI area in imaging experiments, were 21.9 ± 4.9 μm (mean ±s.d.) and 15.9 ± 2.6 μm, respectively (*Figure 4C*, p<0.001 Wilcoxon Rank Test). Within each genotype, soma

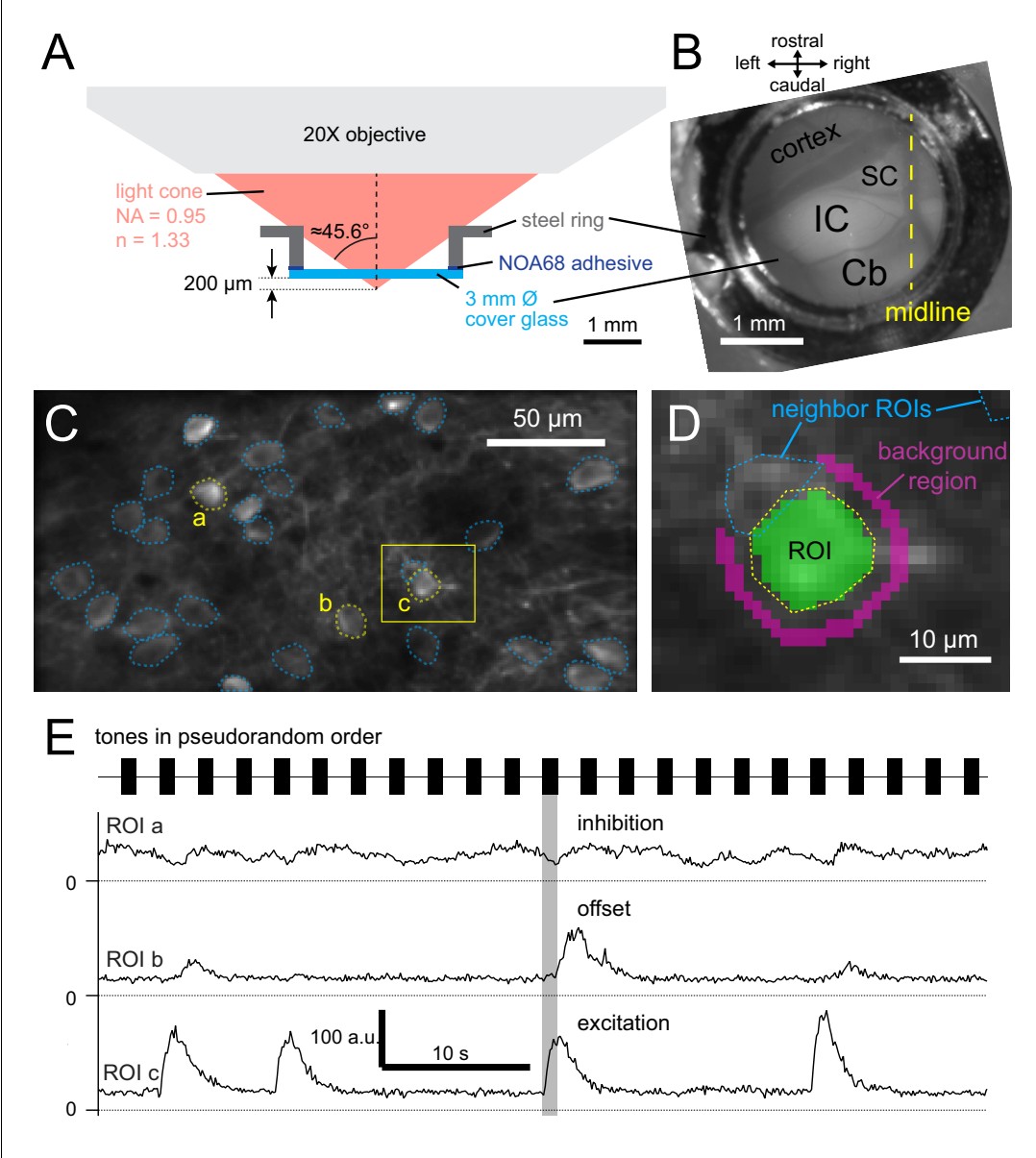

**Figure 2.** Calcium imaging of dorsal inferior colliculus in awake mice. (**A**) Illustration of the cranial window construct imaged with a numerical aperture (NA) 0.95 water-immersion objective. (**B**) Top-down view of cranial window, showing optically exposed inferior colliculus (IC), superior colliculus (SC), cerebral cortex (cortex) and cerebellum (Cb). (**C**) Averaged GCaMP6s fluorescence in a 256 × 128 μm area, showing regions-of-interest (ROIs; dotted lines) defined around neuronal somata. (**D**) Enlarged view from C, showing pixels included in an ROI (green overlay) and a surrounding 2 μm wide background region (magenta overlay). Pixels belonging to other ROIs were excluded from the background region. (**E**) Background-subtracted fluorescence over time of the ROIs labeled a,b,c in C. Tones of 1 s duration with different frequencies and intensities were played, evoking different responses in IC neurons. The same sound (shaded area) evoked an inhibitory response in ROI a, an offset response in ROI b and a sustained excitatory response in ROI c.

DOI: https://doi.org/10.7554/eLife.49091.005

The following figure supplement is available for figure 2:

**Figure supplement 1.** Reduction of scanner noise by sinusoidal galvo scanning.

DOI: https://doi.org/10.7554/eLife.49091.006

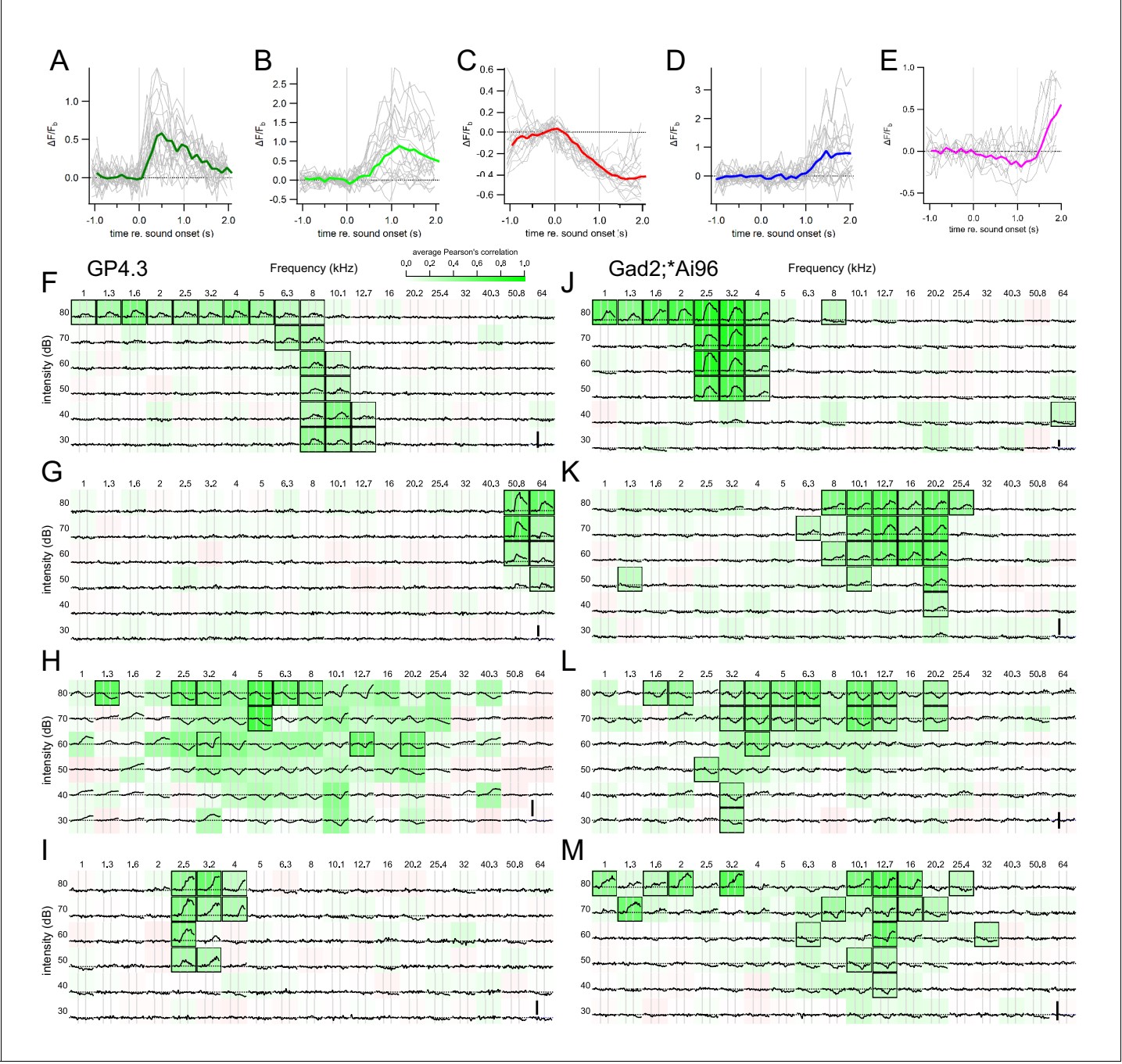

**Figure 3.** Different sound evoked responses and frequency response areas of representative example cells. (A) An onset fluorescence response to a 1 s pure tone. Gray vertical lines indicate onset and offset of sound stimulus. Colored and gray traces are average and individual trial fluorescence changes, respectively, normalized to a one-second baseline immediately before the stimulus onset ($\Delta F/F_b$). (B) A sustained fluorescence response. (C) An inhibitory response. (D) An offset response. (E) A mixed inhibitory and offset response. Example FRAs from GP4.3 (F–I) and Gad2;Ai96 (J–M) mice. Each subplot shows the average $\Delta F/F_b$ to a stimulus of the specified frequency and intensity. Background color shows average Pearson's correlation among repetitions, indicating consistency of response (*Geis et al., 2011*). Black squares mark significant correlation from bootstrap analysis. Cells in (G) and (J) are examples of cells with an onset FRA; cells in (F) and (K) showed a sustained FRA; cells in (H) and (L) were inhibited by sound; cells in (I) and (M) showed a mixture of different response classes. (I) A typical onset-offset cell with frequency-dependent responses: onset fluorescence responses to 2.5 kHz tones and offset fluorescent responses to 3.2 and 4 kHz tones. (M) A cell showing intensity-dependent responses: at 12.7 kHz, low intensity tones evoked a decrease in fluorescence (inhibited) while tones at 80 dB elicited an offset response. Tones at 60–70 dB elicited a mixture of inhibition and offset responses. Vertical scale bars in (F–I) indicates 1 $F_b$.

DOI: https://doi.org/10.7554/eLife.49091.007

*Figure 3 continued on next page*

*Figure 3 continued*

The following source data and figure supplements are available for figure 3:

**Source data 1.** Fluorescence kinetics source data CSV file containing fluorescence kinetics of ROIs, genotype of animal and type of FRA.
DOI: https://doi.org/10.7554/eLife.49091.010
**Figure supplement 1.** Kinetics of fluorescence responses.
DOI: https://doi.org/10.7554/eLife.49091.008
**Figure supplement 2.** Another example FRA from a GP4.3 mouse, showing onset response and broad tuning.
DOI: https://doi.org/10.7554/eLife.49091.009

size did not differ significantly among cells with different FRA classes. For GP4.3: onset cells 12.3 ± 2.6 µm, sustained 12.5 ± 2.0 µm, inhibited 13.0 ± 1.8 µm, offset 12.3 ± 2.0 µm (p=0.112, Kruskal-Wallis Rank test). For Gad2;Ai96: onset cells 16.6 ± 3.4 µm, sustained 17.4 ± 3.6 µm, inhibited 18.1 ± 3.3 µm, offset 15.6 ± 3.1 µm (*Figure 4C*; p=0.033, Kruskal-Wallis Rank test, followed by Dunn's [Dunn-Holland-Wolfe] test where none of the comparisons reached critical value; only cells with non-mixed FRAs were compared because of the heterogeneity and small sample size of the different mixed response classes).

## Detailed kinetics of the different response classes

The temporal kinetics of the $F_{GCaMP}$ response of each cell were assessed by averaging the fluorescence change across all stimuli that showed a significant response. Single exponential functions were fit to the onset (0–1 s re stimulus onset) and the offset/decay (0.5–4 s re stimulus offset) periods (*Figure 3—figure supplement 1A,B*). Time constants were restricted to be positive. Due to the limited stimulus duration, any fit resulting in an onset time constant ($\tau_{onset}$) greater than 1000 ms to an increase in fluorescence was considered sustained activity and these fit constants were not used for averaging. For excitatory responses (onset, sustained and offset), a time constant of around 1 s (*Figure 3—figure supplement 1D*) was found for the decay period, similar to the decay time constant reported previously for GCaMP6s (*Chen et al., 2013*).

## Electrophysiological correlates of response classes

To relate the different fluorescence responses to spiking patterns, which are highly heterogeneous within the IC (e.g. *Willott et al., 1988a*; *Willott et al., 1988b*; *Tan and Borst, 2007*), we combined $Ca^{2+}$ imaging with *in vivo* juxtacellular recordings in awake mice. We recorded from neurons showing onset, sustained and inhibitory tone-evoked fluorescence responses. For fluorescence responses that we classified as onset, electrophysiological recordings showed that this class can represent a cell whose firing rate quickly adapted during our one-second long stimuli (example in *Figure 5A–C, D* left). We observed sustained fluorescence responses that corresponded to sustained firing (*Figure 5D* middle) with different amount of adaptation. Inhibitory responses corresponded to cells that reduced their spontaneous firing upon sound stimulation (*Figure 5D* right).

We further characterized the relationship between fluorescence responses and firing patterns by fitting a model to our data set (*Figure 5E,F*, *Figure 5—figure supplements 1–3*; see Materials and methods for equations). We found that on average, an action potential led to a median increase of 0.35 (GP4.3; n = 5 cells) or 0.14 (Gad2;Ai96; n = 12 cells) times the minimal fluorescence ($F_0$), with a median zero-to-peak rise time of 480 ms (GP4.3) or 560 ms (Gad2;Ai96) and a median decay time constant of 1.04 s (GP4.3) or 1.36 s (Gad2;Ai96). The model could explain between 50% and 95% of the variance in the fluorescence (median: 79%; average ±s.d.: 77 ± 13%).

## Spatial distribution of frequency tuning

The widespread and homogeneous expression of GCaMP6s in the IC of the transgenic mice allowed a good overview of its functional organization. We aligned the position of the cells from multiple animals (GP4.3: n = 7 mice; Gad2;Ai96: n = 6 mice) by anatomical landmarks (midline, anterior and posterior extent of the exposed IC, lateral extent of the exposed IC), and plotted the cells on this common anatomical coordinate system (*Figure 6A*; *Figure 6—video 1*).

Interestingly, we observed a central strip running in the caudomedial-rostrolateral orientation of cells responding to lower frequencies, while CF progressively increased in both the caudolateral and

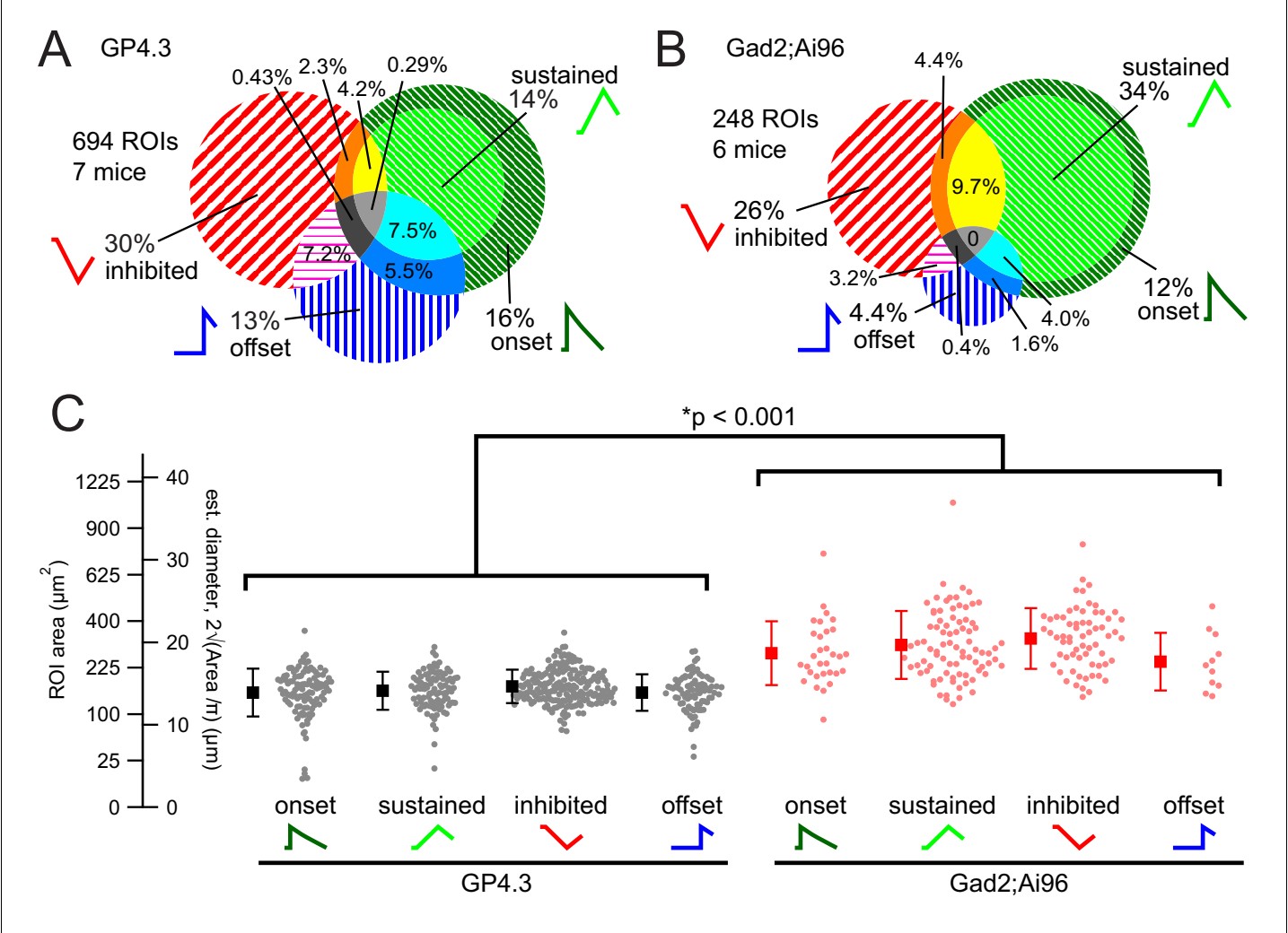

**Figure 4.** Proportion of response classes and relationship with cell size. (**A–B**) Proportions of response classes in GP4.3 (**A**) and GAD2;Ai96 animals (**B**). (**C**) Bee swarm plot and mean ±s.d. for cell size for ROIs of different non-mixed response classes, separated by genotypes, measured as the size of the ROI (*Area*) in imaging experiments. Estimated diameters (d) were calculated by assuming a circular shape (i.e. $\log(\mathrm{CF}) = \sum_{i=0}^{4} a_i r^i$). Cells imaged in Gad2; Ai96 mice were on average larger than those in GP4.3 mice.

DOI: https://doi.org/10.7554/eLife.49091.011

The following source data is available for figure 4:

**Source data 1.** IC ROI Area source data CSV file containing fluorescence area and estimated diameter of ROI, genotype of animal and type of FRA.
DOI: https://doi.org/10.7554/eLife.49091.012

the rostromedial direction. We tried to estimate the orientation of the spatial organization by projecting the x,y-coordinates of all neurons onto an axis with a parametrized angle θ, while simultaneously fitting the log-transformed CF using a polynomial model (see Materials and methods). We used a polynomial model fit as a way to extract any general direction along which CFs may diverge, without imposing any presumption about the spatial CF dependence. We found that a fourth order polynomial captured the variance maximally; further increasing its order did not lead to better fits. The results of the fit are presented as contour lines in *Figure 6A*, with the best orientation (θ) 50° from the medial-lateral (x) axis. A plot along this orientation shows a high fraction of low CF (~4 kHz) neurons near the 625 µm position, with the mode going towards ~20 kHz at the 1300 µm end (*Figure 6B*). The increase in CFs on the rostromedial side was not as pronounced, likely due to the low number of neurons sampled in this region. The solid line in *Figure 6B* shows the best polynomial

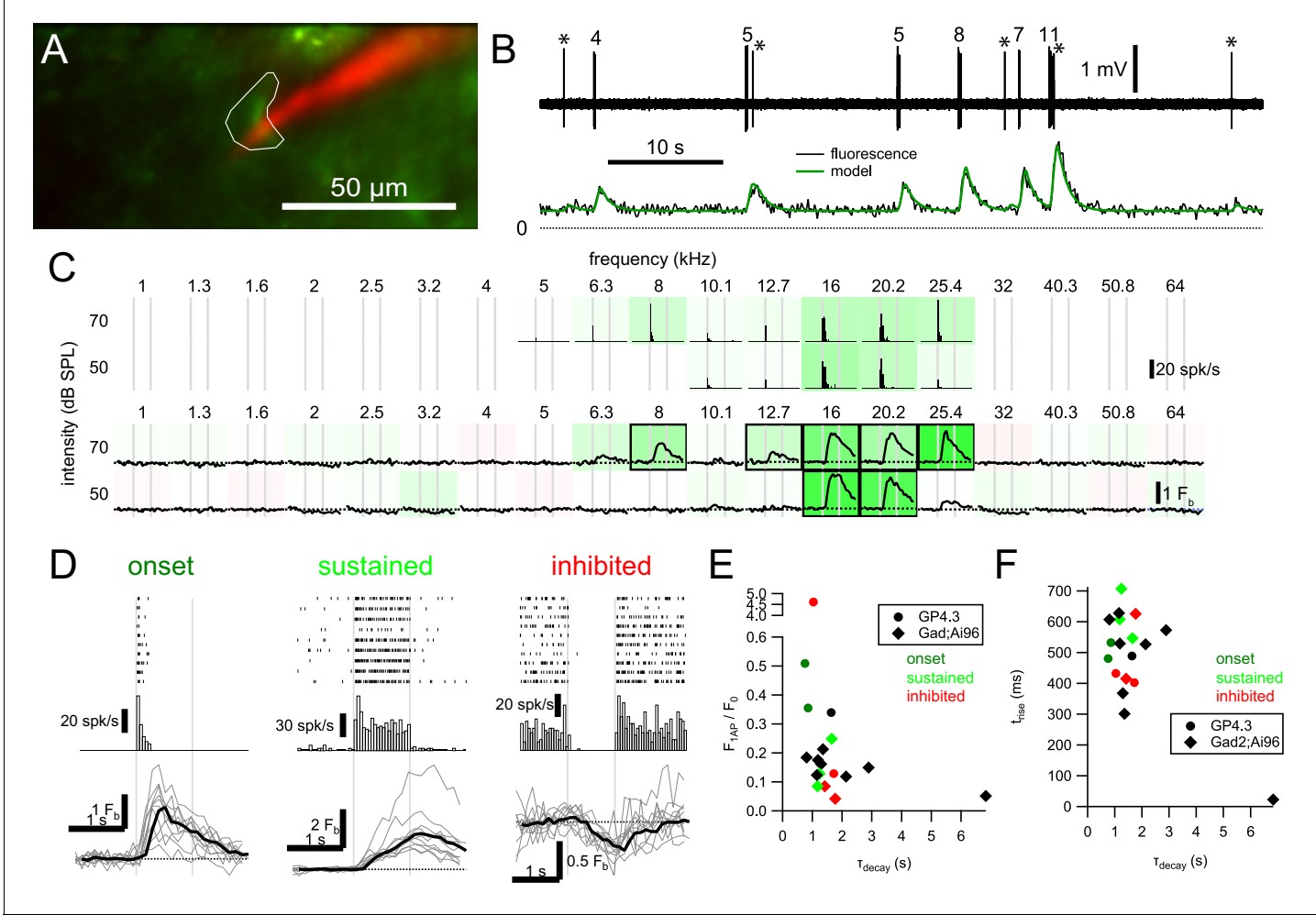

**Figure 5.** Relationship between GCaMP6s fluorescence and spikes in IC neurons. (**A**) Imaging of a GCaMP6s+ cell (green) in an awake GP4.3 animal with simultaneous juxtacellular recording with a pipette filled with Alexa 594 (red). (**B**) Spiking pattern (upper) and fluorescence (lower) of the example cell can be well related by a linear convolution of spike rate with a linear ramp, exponential decay kernel (model: green). Fit parameters: $F_0$ = 28.8 a.u.; $\Delta F_{1AP}$ = 10.2 a.u.; $t_{rise}$ = 532 ms; $\tau_{decay}$ = 859 ms. Numbers above bursts indicate number of spikes, and asterisks (*) mark single spikes. (**C**) Frequency tuning of the cell. Peristimulus time histogram (PSTH; upper, 50 ms bins, 10 repetitions) and mean fluorescence change of the cell in response to 1 s tone bursts. (**D**) Raster plots, PSTHs and fluorescence responses (mean: black; individual: gray) of an onset cell (same example as A-C), a sustained cell and an inhibited cell. Due to their rarity, we have not obtained a simultaneous recording for offset cells. (**E**) Relation between normalized single action potential amplitude ($F_{1AP}/F_0$) and decay time constant ($\tau_{decay}$). (**F**) Relation between rise time (zero-to-peak; $t_{rise}$) and $\tau_{decay}$. Color in E and F indicate response class (red: inhibited; dark green: onset; light green: sustained; black: no tone-evoked response) of the cell.

DOI: https://doi.org/10.7554/eLife.49091.013

The following source data and figure supplements are available for figure 5:

**Source data 1.** Ground-truth model fitting source data CSV file containing model parameters from ground-truth data fitting and the variance explained, genotype of animal and type of FRA.
DOI: https://doi.org/10.7554/eLife.49091.017
**Figure supplement 1.** Example model fit to a cell with a sustained FRA.
DOI: https://doi.org/10.7554/eLife.49091.014
**Figure supplement 2.** Example model fit to a cell with an inhibited FRA.
DOI: https://doi.org/10.7554/eLife.49091.015
**Figure supplement 3.** Example model fit to spontaneous activity of a cell.
DOI: https://doi.org/10.7554/eLife.49091.016

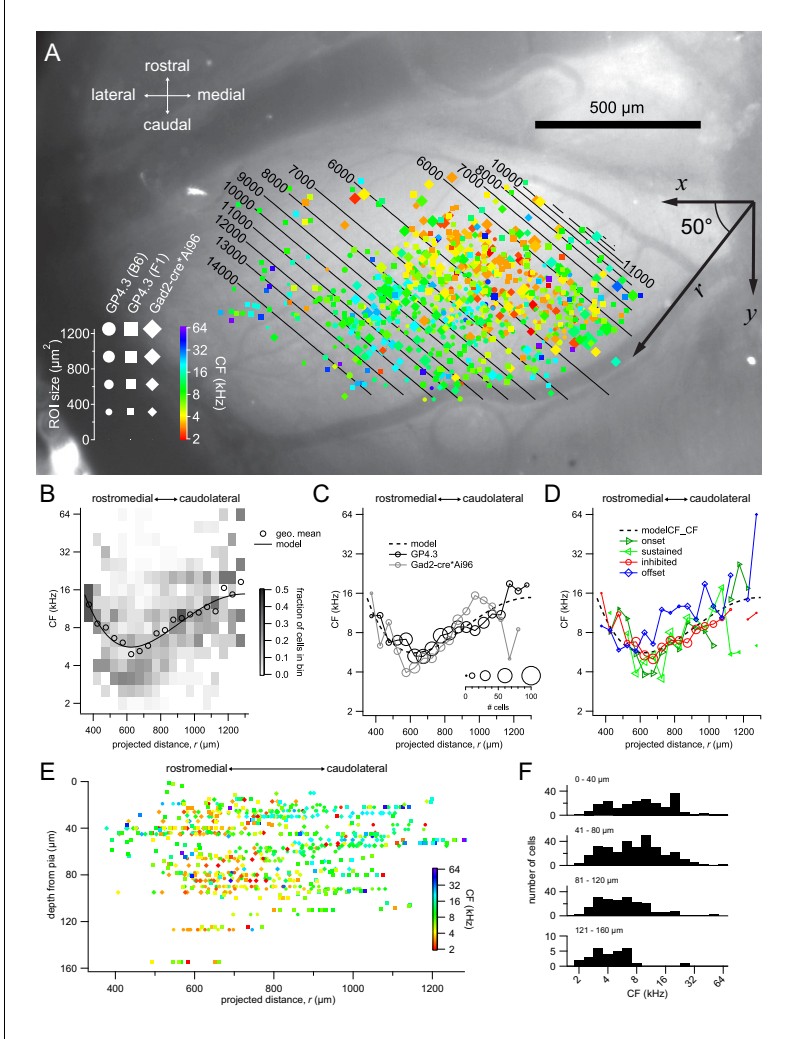

**Figure 6.** Tonotopic organization. (**A**) Combined spatial distribution of characteristic frequencies (CFs) in the two transgenic lines, aligned to the same top-down image of an exposed left IC. Symbol size represents the size of the ROI, while colors indicate different CFs. Cells from Gad2;Ai96 mice were marked by diamonds. For GP4.3, the shape of the symbol represents ROIs from C57BL/6J (circles) or B6CBAF1 (squares) background. To capture a direction of tonotopy, CF was fitted with a $4^{th}$-order polynomial $r = x \cos\theta + y \sin\theta$ where $r$ is a presumed direction of tonotopy at an angle $\theta$ (i.e. $r = x\cos\theta + y\sin\theta$), with the origin being the contact point between superior and inferior colliculi at the midline. The best fit yielded an angle $\theta$ of 50° from the medial-lateral axis. Contour lines indicate the predicted CF of the fit. (**B**) Geometric mean (open circles) and probability distribution (background shading) of CF along the 50° line, in 75 μm bins. Black trace shows the fitted $4^{th}$-order polynomial. (**C**) Geometric mean of CF along the 50° line for the GP4.3 (black) and Gad2-cre*Ai96 (gray) lines. Size of symbol represent number of cells in each 75 μm bin. (**D**) Geometric mean of CF for cells of onset, sustained, inhibited or offset FRAs. (**E**) Depth dependence of CFs along the 50° line. (**F**) Histogram of CF grouped according to depth from pia surface. Note that low CF cells are present at all imaging depths.

DOI: https://doi.org/10.7554/eLife.49091.018

The following video, source data, and figure supplements are available for figure 6:

**Source data 1.** IC CF Distribution source data CSV file containing 3D coordinate of ROIs, projected distance of ROIs, genotype of animal and type of FRA.
DOI: https://doi.org/10.7554/eLife.49091.027

**Figure supplement 1.** Tonotopic organization of a single GP4.3 animal (Mouse 20976–07; B6CBAF1/J background), registered to widefield image in *Figure 6A*.
DOI: https://doi.org/10.7554/eLife.49091.019

**Figure supplement 2.** Tonotopic organization of a single GP4.3 animal (Mouse 11605–01; C57BL/6J background), registered to widefield image in *Figure 6A*.
DOI: https://doi.org/10.7554/eLife.49091.020

**Figure supplement 3.** Tonotopic organization of a single Gad2;Ai96 animal (Mouse 12156–03), registered to widefield image in *Figure 6A*.
DOI: https://doi.org/10.7554/eLife.49091.021

*Figure 6 continued on next page*

*Figure 6 continued*

**Figure supplement 4.** Tonotopic organization of a single Gad2;Ai96 animal (Mouse 12156–04), registered to widefield image in *Figure 6A*.
DOI: https://doi.org/10.7554/eLife.49091.022
**Figure supplement 5.** Tonotopic organization of a single GP4.3 animal (Mouse 11605–04; C57BL/6J background), registered to widefield image in *Figure 6A*.
DOI: https://doi.org/10.7554/eLife.49091.023
**Figure supplement 6.** Tonotopic organization of a single Gad2;Ai96 animal (Mouse 14234–01), registered to widefield image in *Figure 6A*.
DOI: https://doi.org/10.7554/eLife.49091.024
**Figure supplement 7.** Relationship between CF and depth of cell.
DOI: https://doi.org/10.7554/eLife.49091.025
**Figure supplement 8.** Tonotopic organization of the most superficial cells.
DOI: https://doi.org/10.7554/eLife.49091.026
**Figure 6—video 1.** Video of all sound-responsive.
DOI: https://doi.org/10.7554/eLife.49091.028

fit, which explained around 13% of the variance in CFs. This tonotopic organization can also be appreciated in individual animals in which a wide extent of the dorsal IC was imaged (*Figure 6—figure supplements 1–3*). Additional examples of the tonotopic organization in individual animals are shown in *Figure 6—figure supplements 4–6*. A similar spatial distribution of CFs was observed for GABAergic and glutamatergic neurons (*Figure 6C*) and for cells showing different response classes (*Figure 6D*).

Based on a clustering analysis, *Barnstedt et al. (2015)* suggested that the central strip of low frequency neurons were from the most dorsal end of the central nucleus, and we did observe a small but significant correlation between depth and log(CF) (*Figure 6—figure supplement 7*; correlation coefficient = $-0.22$; n = 799; p=$3 \times 10^{-10}$). However, a plot of CF along the best orientation from the polynomial fit as a function of depth illustrates that most cells were located within 100 µm from the surface, and that the low frequency neurons were also among the most superficial cells (*Figure 6E*). When we limited the fit to neurons < 50 µm from the pia surface, the same tonotopic pattern appeared (*Figure 6—figure supplement 8*). There was an over-representation of low CF neurons in the deepest regions (121–160 µm deep, *Figure 6E,F*). We attribute this to a sampling bias in which the low frequency region coincided with the center of the cranial window, where it was possible to image somewhat deeper than towards the edges (*Figure 6E*). Objects in the center of the window profit from utilization of the full NA for focusing excitation light, which is especially critical for two-photon excitation, and in the center both excitation and fluorescence emission light is least obstructed (*Figure 2A*). Restricting the analysis to cells in the central strip (525–725 µm of projected area) reduced the correlation coefficient for the relation between depth and log(CF), but it remained significant (*Figure 6—figure supplement 7*; correlation coefficient = $-0.17$, n = 295; p=0.003), hinting at differences between deeper and the most superficial cells. While we do not exclude the possibility of imaging into the central nucleus of the IC, we believe the observed tonotopy to be a good representation of the IC shell region because all CFs were represented among the most superficial neurons (*Figure 6E,F*).

## Spatial distribution for response classes

We next asked whether there was any obvious spatial organization of the different response classes. *Figure 7A* shows the proportion of different response classes along our presumed tonotopic axis, which looked rather homogenous, without any obvious difference between cells rostromedial and caudolateral to the CF minimum. The same pattern holds for the orthogonal direction (*Figure 7B*).

## Neurons with movement related activity and their localization

Interestingly, we observed cells that showed fluorescence transients in the absence of sound presentation (*Figure 8A,B*). Numerous studies have shown both ascending and descending somatosensory projections to the IC (*Aitkin et al., 1981*; *Künzle, 1998*; *Jain and Shore, 2006*; *Zhou and Shore, 2006*; *Lesicko et al., 2016*; *Patel et al., 2017*), as well as from nuclei that are upstream from IC such as the dorsal cochlear nucleus, which also receive somatosensory inputs (*Wu et al., 2014*). In addition, the IC receives inputs from the motor cortex (*Cooper and Young, 1976*; *Olthof et al.,*

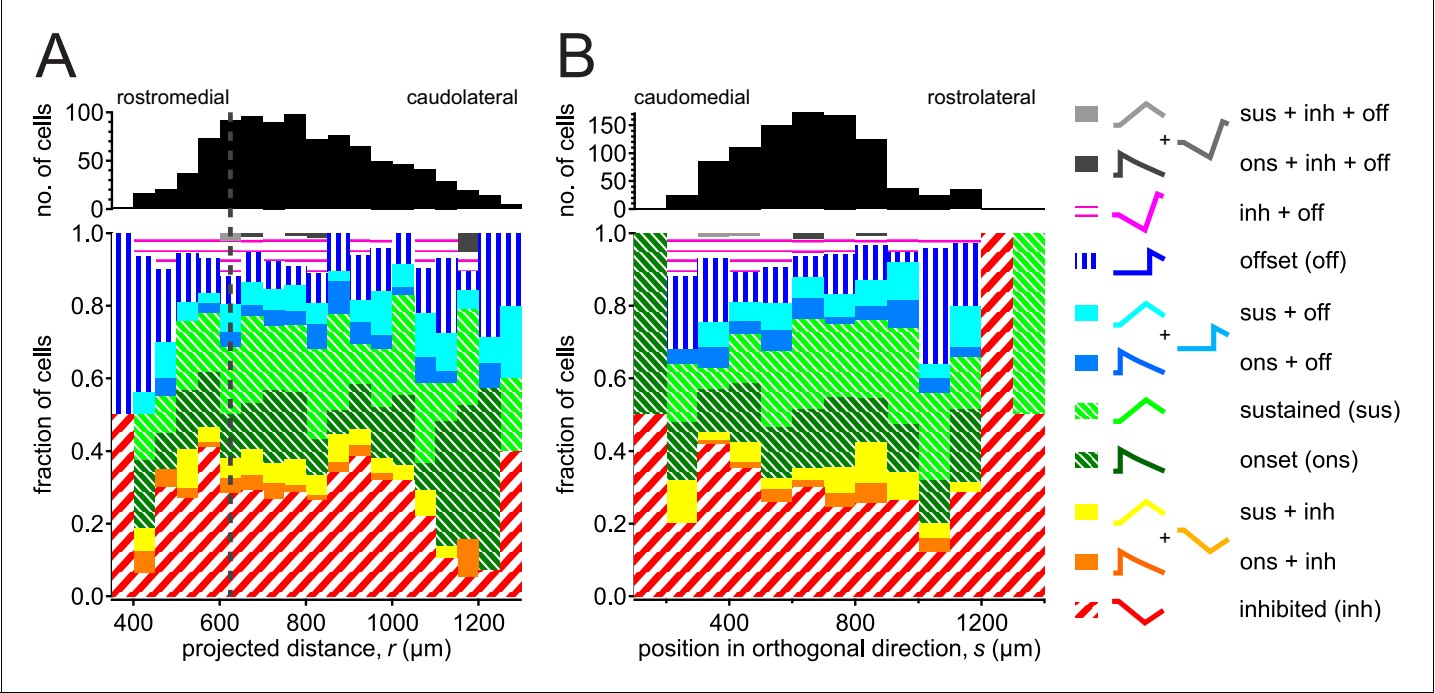

**Figure 7.** Spatial organization of response classes. Similar proportions of response classes along (**A**) the presumed tonotopic axis (bin size 50 µm) and (**B**) the direction orthogonal to the axis (bin size 100 µm). Zero position was taken as the contact point between superior and inferior colliculi at the midline. Vertical dashed line in (**A**) denotes the minimal CF position based on our polynomial fit (625 µm).

DOI: https://doi.org/10.7554/eLife.49091.029

The following source data is available for figure 7:

**Source data 1.** IC Response Distribution source data CSV file containing 3D coordinate of ROIs, projected distance of ROIs, genotype of animal and type of FRA.

DOI: https://doi.org/10.7554/eLife.49091.030

*2019*). This prompted us to investigate whether these 'spontaneous' activities could be attributed to non-auditory inputs. Our recordings were performed while the animal was passively awake, which allowed us to observe and correlate the voluntary movement of the animal to simultaneously recorded calcium transients.

We found that the onset of many of these spontaneous transients coincided with movement events of the animal (paw and facial movements in *Figure 8A*). By correlating movement and fluorescence, we detected 165 (out of 1359) cells that showed a positive correlation ($r > 0.25$) with facial movements during the spontaneous recording period. To exclude potential false-positive detection due to motion artefacts (i.e. cells moving into and out of focus), we excluded cells with fluorescence transients that did not show the typical ~1 s exponential decay kinetics, which last much longer than the brief image shifts that could accompany movement events (image shifts in *Figure 8A*, *Figure 8— figure supplement 1*). Since animal movement inevitably produces sound that may activate the IC neurons, leading to an apparent movement sensitivity, we also excluded the cells that showed excitatory or offset responses in their FRA. In the end, 41 cells showed spontaneous calcium transients that correlated with animal movement and that could not be explained by their FRA, which was either inhibited-type or showed no clear pure tone evoked responses (example FRAs in *Figure 8F*). These cells seemed to be enriched at the caudolateral side of the dorsal IC (*Figure 8C–D*). Four of the 41 cells were found in the Gad2;Ai96 line, in agreement with a recent report that inputs from motor or somatosensory cortex can directly target GABAergic cells (*Olthof et al., 2019*), and the remainder in the GP4.3 line, making them 1% of GABAergic and 4% of investigated glutamatergic cells, respectively.

## Comparison of tonotopic organization with histological data and literature

*Figure 9A and E* show two relatively superficial brain sections (within 80 µm and 120 µm from dorsal surface, respectively) from one Gad2;Ai96 and one GP4.3 animal stained for GAD67 after two photon imaging. We overlaid the line representing the neurons with the lowest CF and compared its location with the border of LCIC and DCIC traced from the latest Allen Reference Atlas (CCFv3) and from the classical reference atlas by *Paxinos and Franklin (2001)*. The minimum frequency line aligns with the demarcation from the Allen Reference Atlas, while that by Paxinos and Franklin lays in the orthogonal direction.

Neurochemical modules with a high density of GAD67-positive terminals have been reported as a hallmark for the LCIC (*Chernock et al., 2004*; *Lesicko et al., 2016*; *Dillingham et al., 2017*). We indeed found modules with higher density of GAD67 terminals in our animals as well (arrowheads in *Figure 9A,E,a* clear example in *Figure 9C*). In the most superficial sections, these modules were not as well defined, but they were contiguous to the more densely stained modules in deeper sections (*Figure 9—figure supplements 1* and *4*; 3D reconstruction in *Figure 9—video 1*), and the superficial modules also had reduced calretinin staining (*Figure 9—figure supplements 2* and *5*), in line with previous findings (*Dillingham et al., 2017*). These modules are the predominant target of the somatosensory projections from the cerebral cortex as well as other brainstem areas (*Lesicko et al., 2016*). We also observed a central strip of dense GABAergic staining (enlarged in *Figure 9B*), which was not contiguous with the modules (*Figure 9—figure supplements 1* and *4*). GCaMP expression in these two example animals are presented for comparison in *Figure 9—figure supplements 3* and *6*.

How do motion-sensitive cells associate with the GAD67-dense modules? *Figure 9D2 and F* show two motion-sensitive neurons retrieved in histology (cells 3 and 4, marked by arrows), both of which resided in close proximity to a GAD67-dense module. We observed that although the somata of both cells seemed to reside in the extramodular regions, both possessed dendrites that extended into a GAD67-dense module (*Figure 10*). *Figure 10A and B* show a dendritic extension of cell 3. For cell 4, a pixel-wise correlation of the *in vivo* somatic fluorescence change (*Junek et al., 2009*) nicely revealed its dendritic arbor at the same focal plane (*Figure 10C,E*). Aligning this to *post hoc* histological staining (*Figure 10D*), we can demonstrate that cell four has dendritic branches extending both inside and outside of a GAD67 module (*Figure 10F*). We suggest that this may be a functional connection scheme for extramodular neurons to enable them to integrate somatosensory or motor with auditory inputs to the IC (*Figure 10G*).

## Discussion

We investigated the functional organization of the dorsal IC in awake mice at the single neuron level. We found that the dorsal IC is tonotopically organized, with isofrequency bands running in a rostro-lateral-caudomedial orientation with frequency tuning varying along the caudolateral-rostromedial direction. The lowest CF band was found in the middle, dividing the dorsal IC into two reversed tonotopic gradients. In the caudolateral part, but not in the rostromedial part, neurons that were sensitive to whisker or other movements were found, which was in agreement with the view that the caudolateral and rostromedial part corresponded to the LCIC and DCIC, respectively. We observed four different types of firing patterns in response to tones, but other than the tonotopical organization, no obvious topographical organization was observed for the firing patterns. GABAergic neurons were on average somewhat larger, but otherwise there were not clear differences with glutamatergic neurons in spatial organization or firing patterns. Our experiments thus provide a functional definition of the major organization principles of the dorsal IC in adult mice.

### Suitability of GP4.3 and GAD2-cre mice for 2P imaging in awake mice

Because of the prominence of GABAergic neurons and their substantial contribution to ascending projections (*Schofield and Beebe, 2019*), we compared the responses and functional organization of GABAergic and glutamatergic neurons in the dorsal IC using *in vivo* two-photon calcium imaging. We confirmed that virtually all neurons expressing GCaMP6s in the GP4.3 line were glutamatergic, but not all glutamatergic neurons were labeled. There may have been a preponderance of labeled

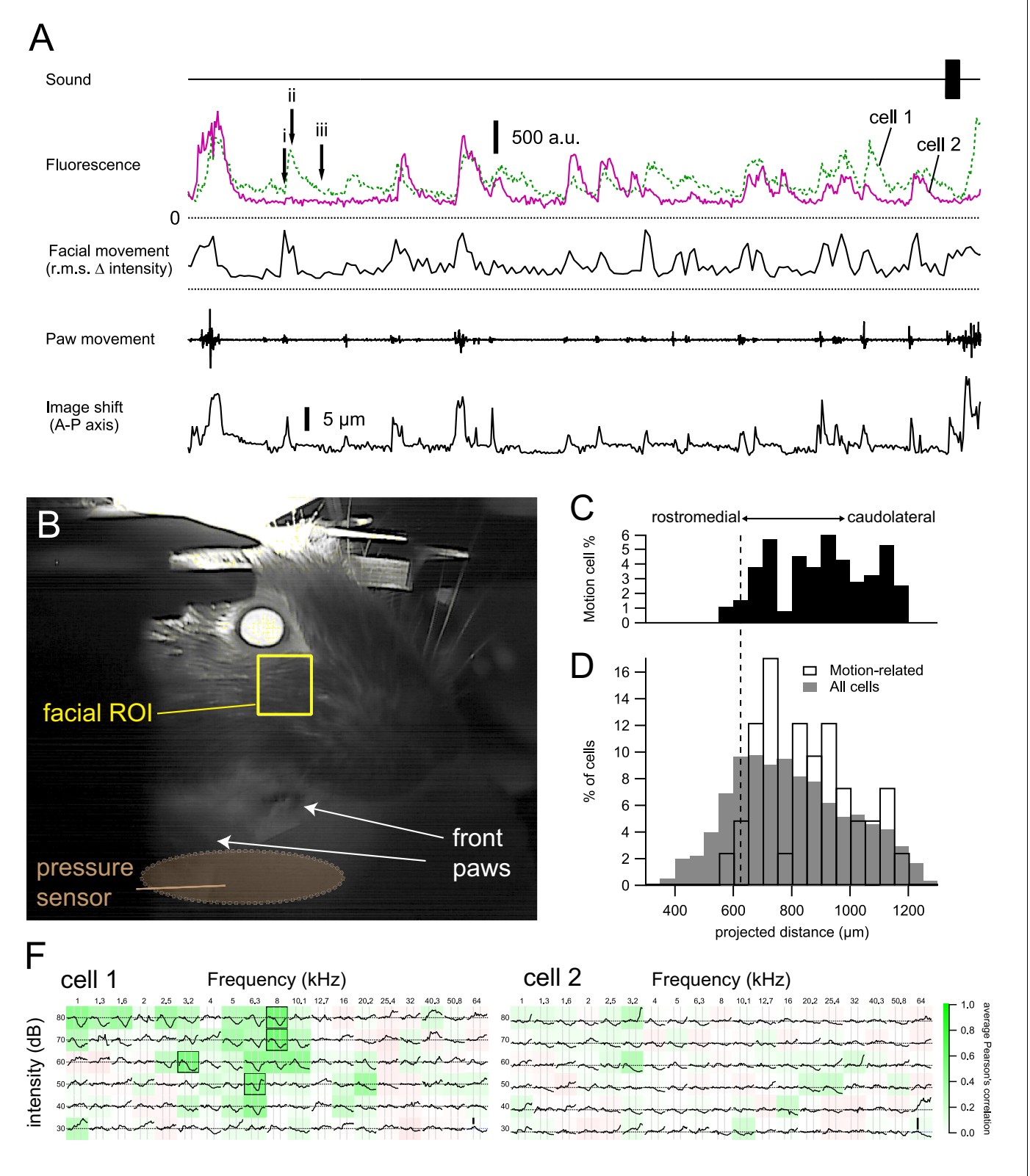

**Figure 8.** Cells with motion-related responses. (**A**) Two example cells that were spontaneously active in the absence of sound presentation. Calcium transients occurred when the animal was moving its paws or showing facial movement (e.g. whisking). These transients decayed much more slowly than motion artefacts (image shifts, bottom trace; see also *Figure 8—figure supplement 1*). (**B**) Facial movements were quantified from a simultaneously recorded video of the animal by calculating the root-mean-square of the changes in pixel intensity between consecutive frames of a rectangular area at

*Figure 8 continued on next page*

*Figure 8 continued*

the whisker pad. (**C–D**) Cells showing motion-related activity were predominently located in the caudolateral part of the dorsal IC. Broken line indicates the minimum CF position obtained from the polynomial fit. (**F**) The motion-related calcium transients were unlikely to be caused by sounds, as the two cells showed either an inhibitory FRA (cell 1) or no clear tone-evoked response (cell 2).

DOI: https://doi.org/10.7554/eLife.49091.031

The following source data and figure supplement are available for figure 8:

**Source data 1.** IC Movement cells source data CSV file containing 3D coordinate of ROIs, projected distance of ROIs, genotype of animal, type of FRA and parameters related to movement-related response.

DOI: https://doi.org/10.7554/eLife.49091.033

**Figure supplement 1.** Evidence that the motion-related responses were not due to motion artefacts in imaging.

DOI: https://doi.org/10.7554/eLife.49091.032

cells around the modules in the LCIC (*Figure 1D*), but a systematic screen of markers would be needed to test for a selective enrichment of glutamatergic subtypes in the GP4.3 line. The large majority of GCaMP6s-positive neurons in the IC of mice from the Gad2;Ai96 line expressed GAD67, in agreement with findings in a related mouse line (*Gay et al., 2018*). In our hands, these two transgenic lines had the advantage that the kinetics of the responses were relatively homogeneous and well approximated by a linear model (*Figure 5B*), suggesting that the expression levels, which are a major determinant of the $Ca^{2+}$ kinetics (e.g. *Éltes et al., 2019*), were similar across cells. Moreover, in contrast to pilot experiments in which we expressed GCaMP6 using adeno-associated viruses, we found little evidence for toxicity. The more sparse expression facilitated the isolation of the responses of individual neurons.

A substantial fraction of cells did not respond to tones. Low probability responses may have been below detection, since the juxtacellular recordings indicated that single APs could not always be detected. Some cells may respond preferentially to more complex sound stimuli, as previously found for neurons in the shell region (*Ehret and Moffat, 1985*; *Aitkin et al., 1994*). As discussed below, some cells may prefer somatosensory or motor stimuli instead of sound stimuli.

## Response types

In both transgenic lines, we observed four different tone-evoked fluorescence response patterns in awake mice: onset, sustained, offset and inhibitory. Our simultaneous juxtacellular recordings showed corresponding patterns. These types were typically also observed in several earlier electrophysiological studies (e.g. *Willott et al., 1988a*; *Willott et al., 1988b*; *Jain and Shore, 2006*; *Xie et al., 2007*). Apart from a possibly lower fraction of GABAergic neurons with pure offset responses (*Figure 4B*), our results are thus in line with a recent study that showed that GABAergic and glutamatergic IC neurons have similar response properties throughout the IC (*Ono et al., 2017*).

The prominence of inhibitory responses in the IC (~44% of sound-responsive neurons) in both transgenic lines was a striking finding in our recordings. An important factor contributing to their prominence is probably that our experiments were done in awake animals, as a recent single unit recording study showed the presence of inhibitory responses as well as higher spontaneous spike rates in IC neurons of awake mice compared to mice under urethane anesthesia (*Duque and Malmierca, 2015*), and synaptic inhibition has also been shown to be prominently present in whole-cell recordings in awake bats (*Xie et al., 2007*).

Offset responses can be found throughout the auditory system including the IC, and they may have an important role in perceptual grouping or in duration discrimination (*Kopp-Scheinpflug et al., 2018*). However, studies in the IC of anesthetized mice, rats and chinchillas showed that offset responses are generally restricted to cells showing band-pass duration tuning, which only responded to much shorter tones than the 1 s employed here (*Chen, 1998*; *Brand et al., 2000*; *Pérez-González et al., 2006*). Another striking finding was therefore that we saw clear evidence for offset responses in 3–11% of sound-responsive cells. A whole-cell study in the mouse IC showed that in most cases offset responses are inherited from upstream, but that they may also be generated de novo as a rebound from inhibition (*Kasai et al., 2012*). Similar to the inhibition class, we therefore suggest that the prominence of offset responses may be related to the increased impact of synaptic inhibition in the IC of awake animals. A study using communication calls suggests

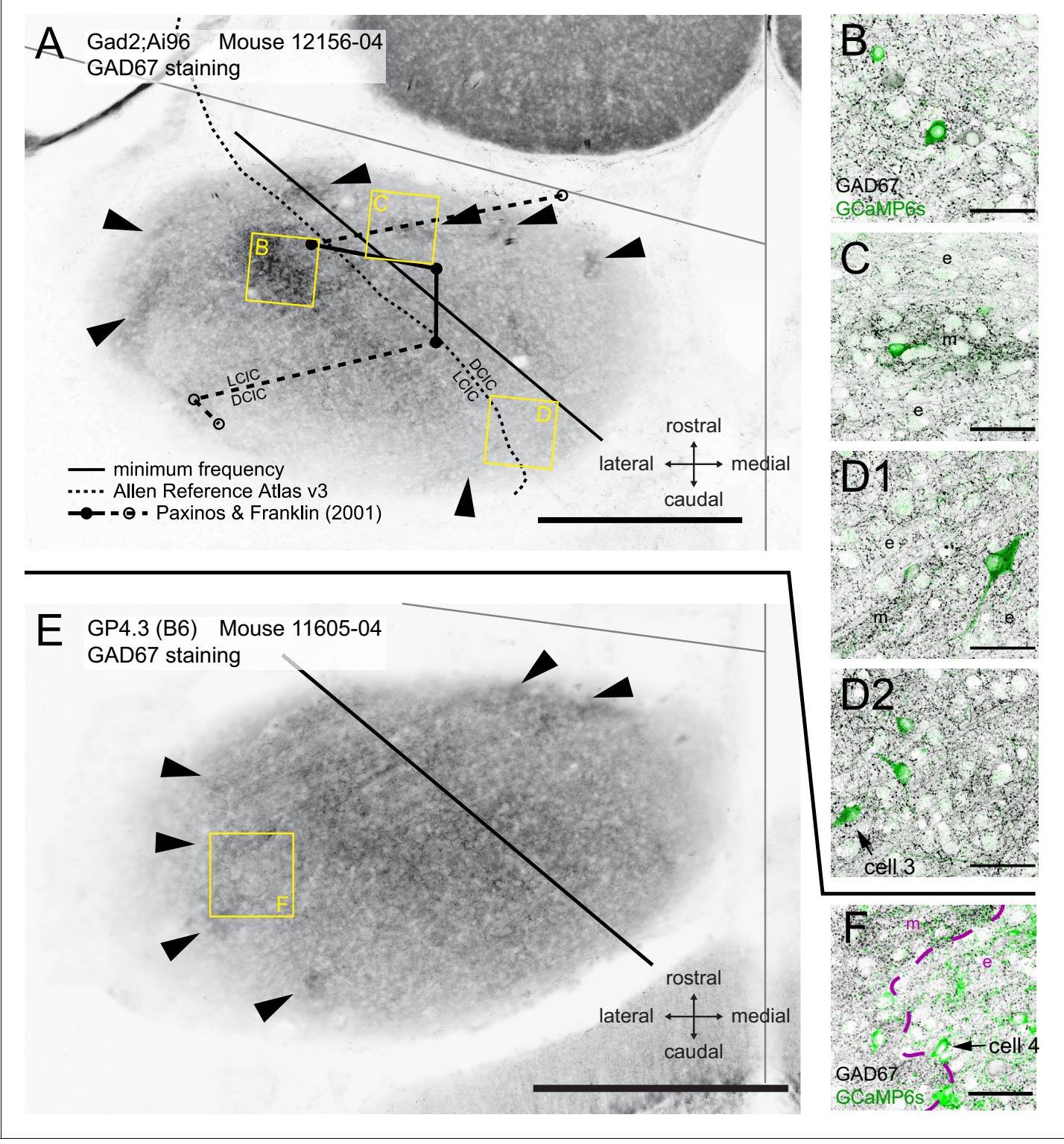

**Figure 9.** Comparison of tonotopic organization with histological data and literature. (**A**) Epifluorescence image of the IC in a horizontal brain section stained for GAD67. This brain slice was from a Gad2;Ai96 mouse after two-photon imaging. Black straight line indicates the minimum frequency location derived from fitting two-photon imaging data. Dashed curve represents the demarcation between the dorsal and the lateral (external) cortices traced from version 3 of the Allen Reference Atlas. Circles connected with solid and broken lines mark the demarcation at the dorsal brain surface traced from the atlas by *Paxinos and Franklin (2001)*. The rostral and caudal end of this demarcation are marked by a dashed line because at the indicated positions on the anterior-posterior axis the whole structure was labeled as LCIC or DCIC, respectively. Modules with dense GAD67 staining,

*Figure 9 continued on next page*

*Figure 9 continued*

considered to be a hallmark for the LCIC, were observed both medially and laterally from each of the three demarcations (arrowheads). We observed a region in the center of the IC whose GAD67 staining density was at least as strong as in the neurochemical modules (square labeled B). (**B–D**) Single confocal optical sections of GAD67 (black) and GCaMP6s (green) staining in different areas of the brain slice corresponding to the yellow squares in **A**. (**B**) Dense GAD67 area in the central region of the IC (<80 µm from dorsal surface). (**C**) An area showing a GAD67-dense module (**m**) in the center, cut transversely and surrounded by extramodular region (**e**) with sparse GAD67 staining. (D1-2) Optical sections at different focal depths of an area showing another GAD67 module from the same slice (m), cut tangentially. Arrow in D2 indicates a cell showing motion-related responses. (**E**) Similar to A but from an imaged GP4.3 mouse. (**F**) Single confocal section for GCaMP6s (green) and GAD67 (black) staining in region marked in E. Arrow marks another cell with motion-related response. Scale bars, A,E: 500 µm; B-D,F: 50 µm.

DOI: https://doi.org/10.7554/eLife.49091.034

The following video and figure supplements are available for figure 9:

**Figure supplement 1.** GAD67 staining in the IC of a series of consecutive 40 µm horizontal brain slices from the same Gad2;Ai96 animal as in *Figure 9A* (animal: 12156–04), displayed from dorsal (top-left) to ventral (right-bottom) showing that the fainter GAD67 staining highlighted in *Figure 9A* is contiguous with the well-stained neurochemical modules in more ventral slices (arrowheads and numbers).

DOI: https://doi.org/10.7554/eLife.49091.035

**Figure supplement 2.** Calretinin staining in the IC of a series of consecutive 40 µm horizontal brain slices from the same Gad2;Ai96 animal as in *Figure 9A* (animal: 12156–04), displayed from dorsal to ventral (left to right, top to bottom).

DOI: https://doi.org/10.7554/eLife.49091.036

**Figure supplement 3.** GFP staining of GCaMP6s in the IC for a series of consecutive 40 µm horizontal brain slices from the same Gad2;Ai96 animal as in *Figure 9A* (animal: 12156–04), displayed from dorsal to ventral (left to right, top to bottom).

DOI: https://doi.org/10.7554/eLife.49091.037

**Figure supplement 4.** GAD67 staining in the IC of a series of consecutive 40 µm horizontal brain slices from the same GP4.3 animal as in *Figure 9E* (animal: 11605–04; C57BL/6J background), displayed from dorsal (top-left) to ventral (right-bottom), showing that the fainter GAD67 staining highlighted in *Figure 9E* is contiguous with the well-stained neurochemical modules in more ventral slices (arrowheads and numbers).

DOI: https://doi.org/10.7554/eLife.49091.038

**Figure supplement 5.** Calretinin staining in the IC of a series of consecutive 40 µm horizontal brain slices from the same GP4.3 animal as in *Figure 9E* (animal: 11605–04; C57BL/6J background), displayed from dorsal to ventral (left to right, top to bottom).

DOI: https://doi.org/10.7554/eLife.49091.039

**Figure supplement 6.** GFP staining of GCaMP6s in the inferior colliculus for the series of consecutive 40 µm horizontal brain slices from the same GP4.3 animal as in *Figure 9E* (animal: 11605–04; C57BL/6J background), displayed from dorsal to ventral (left to right, top to bottom).

DOI: https://doi.org/10.7554/eLife.49091.040

**Figure 9—video 1.** Video showing the 3D reconstruction of the most dorsal aspect of the left and right inferior colliculi in animal 12156–04 from 40 µm serial sections.

DOI: https://doi.org/10.7554/eLife.49091.041

that offset responses may be even more prominent following more complex sound stimuli, such as communication calls (*Akimov et al., 2017*), possibly by a summation of inhibition at different frequency bands (*Sanchez et al., 2008*).

## Tonotopical organization of the dorsal IC

Following pooling of the data from several experiments, we observed a tonotopic gradient that ran along a gradient from caudolateral to rostromedial. The gradient reversed at the site where the CFs were the lowest. This reversal is in line with the banding patterns observed with epifluorescent calcium imaging in unanesthetized mice before and at hearing onset; the pre-hearing spontaneous neural activities originate from the cochlea, thus regions activated synchronously likely receive input from the same tonotopic area (*Babola et al., 2018*). In addition, they observed a double band pattern with higher frequency stimulation in the juvenile (p15) mice. Interestingly, our data are also compatible with two earlier two-photon calcium imaging studies (*Ito et al., 2014*; *Barnstedt et al., 2015*). *Ito et al. (2014)* reported gradients in the medial regions of the dorsal IC, which ran from lateral (low-frequency) to medial (high-frequency), whereas *Barnstedt et al. (2015)* reported gradients in a mediorostral to laterocaudal direction.

We found considerable variability in CFs within a band. Imprecisions in the alignment of imaging areas and variability between animals may have contributed to the variability, but substantial variability was also observed within a single animal, in line with previous results (*Ito et al., 2014*; *Barnstedt et al., 2015*). Two-photon calcium imaging studies in the mouse auditory cortex have met with variable amount of microheterogeneity, apparently depending on layer, calcium dye, or the use

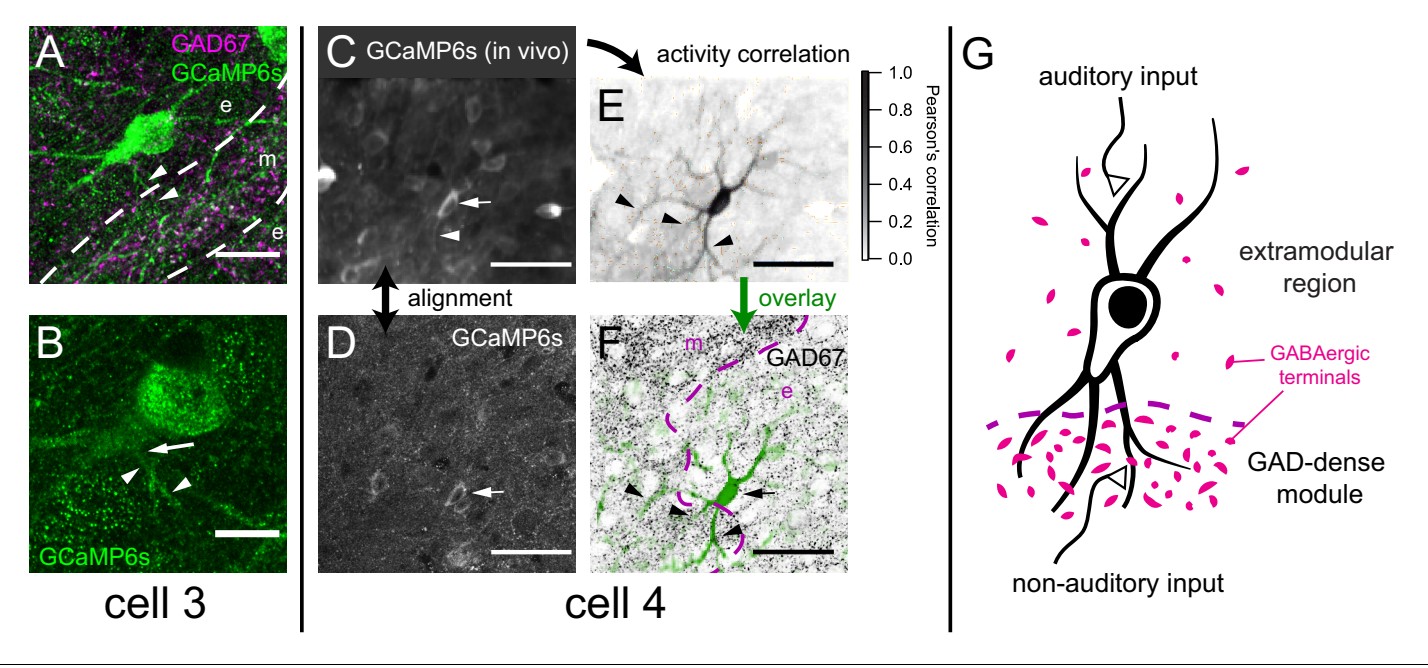

**Figure 10.** Example motion-sensitive cells with dendritic arbor extending into GAD67-dense modules. (**A**) Maximum projection of GCaMP staining showing dendritic arbor of cell three in *Figure 9D2*, overlaid with a single optical section of GAD67 staining showing a module (m; at same focus as *Figure 9D2*). While its soma is extramodular (e), at least one branch of its dendrite (arrowheads) appeared to extend into the modular region. (**B**) A single confocal section of cell three showing the root (arrow) of the dendrite labeled in A (small arrowheads). (**C**) Averaged GCaMP6s fluorescence for the *in vivo* session imaging area in *Figure 9F*, revealing a dendrite of cell four extending into the modular region. (**D**) Single confocal section of a fixed brain slice stained for GCaMP6s. (**E**) Dendritic arbor of cell four was revealed using pixel-wise correlation to the average somatic fluorescence (*Junek et al., 2009*), showing extension into the modular region (arrowheads). (**F**) Background subtracted pixel correlation from E (green) overlaid onto GAD67 staining (black). (**G**) Schematic representation of hypothesis that integration of auditory and non-auditory inputs by multisensory neurons in the IC can be based on extension of their dendrites into both modular and extramodular regions. Scale bars: A: 20 µm; B: 10 µm; C-F: 50 µm.
DOI: https://doi.org/10.7554/eLife.49091.042

of anesthetics (*Bandyopadhyay et al., 2010*; *Rothschild et al., 2010*; *Winkowski and Kanold, 2013*; *Issa et al., 2014*; *Kato et al., 2017*; *Tischbirek et al., 2019*).

The four response types showed a similar tonotopic organization, but considerable microheterogeneity. This microheterogeneity was larger than in an earlier study in which bulk loading of an organic Ca indicator was used (*Ito et al., 2014*), which may have made it more difficult to isolate responses from individual cells. Our results are in line with patch-clamp results showing considerable heterogeneity between adjacent cells within the dorsal IC, which extended to heterogeneity in their inputs (*Geis et al., 2011*). The microheterogeneity was similar for GABAergic and glutamatergic neurons. Our results thus differ from auditory cortex, where response patterns are different and local microheterogeneity is much smaller for (parvalbumin-positive) GABAergic neurons than for glutamatergic neurons (*Maor et al., 2016*; *Liang et al., 2018*; *Liu et al., 2019*). We conclude that despite the shared input and firing behavior during development (*Babola et al., 2018*), functional heterogeneity dominates within bands in the dorsal IC of adult mice. Whereas some of the mechanisms that underlie the microheterogeneity within the auditory cortex are being elucidated (*Kato et al., 2017*; *Tao et al., 2017*; *Vasquez-Lopez et al., 2017*), they remain to be explored for the dorsal IC.

## Presence of non-auditory inputs in LCIC

We monitored spontaneous whisker and general body movements, and used strict criteria to exclude movement artefacts. As we restricted the analysis to neurons that were not excited by tones, we consider it unlikely that the observed movement-related responses were caused by self-generated sounds. Despite these severe restrictions, we did find that a few percent of the cells were excited during whisker or body movements, suggesting a substantial role for somatosensory or

motor inputs. Little is known about the functional characteristics of motor inputs that are not related to eye movements (*Gruters and Groh, 2012*). For somatosensory inputs, the dominant effect appears to be inhibitory and only a minority of cells have been shown to respond to unimodal tactile stimuli (*Aitkin et al., 1978*; *Aitkin et al., 1981*; *Zhou and Shore, 2006*), which would not help detection based on our selection criteria. We did not further discriminate between different movements as they tended to be heavily correlated. Moreover, it is known that neurons in the LCIC have broad tuning for somatosensory input (*Aitkin et al., 1981*). Movement-sensitive neurons were not found in the high-frequency DCIC regions, but only in the putative CNIC region and the LCIC.

A defining feature of the LCIC is the presence of neurochemical modules, which have been shown in many species, including rats (*Chernock et al., 2004*; *Choy Buentello et al., 2015*), but also in both adult (*Choy Buentello et al., 2015*; *Lesicko et al., 2016*; *Patel et al., 2017*) and developing mice (*Dillingham et al., 2017*). While it is still unclear how neurons in these modules and extramodular zones connect with each other, there is evidence that somatosensory inputs project predominantly to the modules, while auditory input projects to the extramodular region (*Lesicko et al., 2016*). The main sources for somatosensory input to the LCIC are the spinal trigeminal nucleus, the dorsal column nuclei and the somatosensory cortex (reviewed in *Gruters and Groh, 2012*). These inputs target the layer two modules, which constitute only a small fraction of the LCIC volume. Moreover, neurons in the modules do not seem to target extramodular IC neurons (*Lesicko and Llano, 2019*). This raises the question how so many cells in the LCIC can be sensitive to non-auditory inputs. We obtained anecdotal evidence that neurons at the border of the modules can extend dendrites into the modules (*Figure 10A,F*), which would allow them to sample non-auditory inputs. Indeed, neurons in LCIC can have extensive dendritic trees (*Smith, 1992*), and Golgi stainings show that cells in layer 3 of LCIC can send dendrites up to layer 1 (*Meininger et al., 1986*; *Malmierca et al., 2011*). The CNIC has few somatosensory inputs, but at least half of the cells in CNIC are affected by stimulation of the dorsal column nuclei (*Gruters and Groh, 2012*). The CNIC may get somatosensory inputs indirectly via the dorsal cochlear nucleus, which is innervated by the spinal trigeminal nucleus and the dorsal column nuclei, or in the form of intracollicular input from the LCIC. Direct stimulation of the different non-auditory inputs would be needed to obtain a better idea of their importance for the observed facial or body movement-related responses.

## Functional parcellation of the dorsal IC

There is little agreement on the precise borders of the different nuclei of the IC, and its parcellation has been different based on whether cell morphology, inputs, or physiological properties were chosen as the main classifier (*Oliver, 2005*). Here we used physiological properties as the main criterion to look at the borders between the DCIC, LCIC and CNIC in the dorsal IC. Our findings suggest that the laterocaudal area with high frequencies is the dorsal edge of the LCIC, whereas the mediorostral part of the dorsal IC belongs to the DCIC. The tonotopic reversal would thus provide a functional demarcation between the dorsal and lateral cortices of the IC. Earlier tracing studies of intracollicular projections in the guinea pig (*Malmierca et al., 1995*) and the rat (*Saldaña and Merchán, 1992*), as well as ascending projections from the cochlear nucleus in the rat (*Malmierca et al., 2002*), have shown interesting V-shaped arrangements in coronal sections, where the medial side corresponds to isofrequency laminae in CNIC and DCIC, while the lateral side corresponds to those in the LCIC. These V-shaped axon terminal plexuses, if extended to the dorsal surface of the IC, provide an anatomical explanation for the observed tonotopic reversal. This is also in agreement with the latest anatomical framework published by the Allen Brain Institute (*Figure 9A*; Allen Mouse Common Coordinate Framework, 2015), previous imaging studies (*Ito et al., 2014*; *Barnstedt et al., 2015*; *Babola et al., 2018*), the known somatosensory inputs to the LCIC, and both the descending and ascending inputs to the IC (*Aitkin et al., 1981*; *Künzle, 1998*; *Jain and Shore, 2006*; *Zhou and Shore, 2006*; *Lesicko et al., 2016*; *Patel et al., 2017*). Surprisingly, we observed GAD67-dense modules in both the medial and the lateral aspect of the IC (arrowheads *Figure 9A,G*), that is within both LCIC and DCIC in either atlas. The medial modules could not be imaged within this study, so their association with somatosensory inputs is currently unclear.

The central strip with neurons tuned to low frequencies in between the LCIC and the DCIC may be the most dorsal extension of the dorsolateral low frequency area of the CNIC (*Stiebler and Ehret, 1985*; *Barnstedt et al., 2015*). This strip contains the large density of GABAergic inputs (*Figure 9A,B*) that is characteristic of the CNIC (*Choy Buentello et al., 2015*). Lemniscal inputs are

known to extend quite dorsally, almost up to the surface, and neurons with short latency sound responses have been found in the dorsal IC, although their exact location was not studied (*Geis and Borst, 2013*). The low tuning in the central strip extended close to the surface, but a more systematic study of their properties would be needed to be able to assign the most superficial layers to either the DCIC or another region. From our data it thus appears that the three main areas of the IC, LCIC, CNIC, DCIC, are readily accessible at the dorsal IC. Their accessibility for imaging studies will thus help to further delineate their functions and the role of their inputs in the future.

# Materials and methods

## Key resources table

| Reagent type (species) or resource | Designation | Source or reference | Identifiers | Additional information |
|---|---|---|---|---|
| Genetic reagent (*Mus musculus*) | B6;129S6-*Gt (ROSA)26Sor*<sup>tm96(CAG-GCaMP6s)Hze</sup>/J | The Jackson Laboratory; originally reported in *Madisen et al. (2015)* | IMSR Cat# JAX:024106, RRID:IMSR_JAX:024106 | Designated as 'Ai96' in this manuscript. Maintained in heterozygous state by backcrossing to C57BL/6J. |
| Genetic reagent (*Mus musculus*) | STOCK Gad2<sup>tm2(cre)Zjh</sup>/J | The Jackson Laboratory; originally reported in *Taniguchi et al. (2011)* | IMSR Cat# JAX:010802, RRID:IMSR_JAX:010802 | Designated as 'Gad2-IRES-Cre' in this manuscript. Maintained in homozygous state after > 10 generations of backcrossing to C57BL/6J, and re-backcrossed to C57BL/6J every 4–5 generations. |
| Genetic reagent (*Mus musculus*) | C57BL/6J-Tg (Thy1-GCaMP6s) GP4.3Dkim/J | The Jackson Laboratory; originally reported in *Dana et al. (2014)* | IMSR Cat# JAX:024275, RRID:IMSR_JAX:024275 | Designated as 'GP4.3' in this manuscript. Maintained in heterozygous state by backcrossing to C57BL/6J, or crossed with CBA/JRj to obtain mice with B6CBAF1/J background. |
| Strain, strain background (*Mus musculus*) | JAX C57BL/6J | Charles Rivers Laboratories | IMSR Cat# JAX:000664, RRID:IMSR_JAX:000664 | |
| Strain, strain background (*Mus musculus*) | CBA/JRj | Janvier Labs | MGI Cat# 6157506, RRID:MGI:6157506 | |
| antibody | chicken anti-GFP (Chicken polyclonal) | Aves | Aves Labs Cat# GFP-1020, RRID:AB_10000240 | IF(1:1000) |
| Antibody | mouse anti-Gad67 (Mouse monoclonal) | Millipore | Millipore Cat# MAB5406, RRID:AB_2278725 | IF(1:1000) |
| Antibody | rabbit anti-NeuN (Rabbit polyclonal) | Millipore | Millipore Cat# ABN78, RRID:AB_10807945 | IF(1:1000) |
| Antibody | mouse anti-parvalbumin (Mouse monoclonal) | Swant | Swant Cat# 235, RRID:AB_10000343 | IF(1:7000) |
| Antibody | rabbit anti-calretinin (Rabbit polyclonal) | Swant | Swant Cat# 7699/4, RRID:AB_2313763 | IF(1:5000) |

*Continued on next page*

*Continued*

| Reagent type (species) or resource | Designation | Source or reference | Identifiers | Additional information |
|---|---|---|---|---|
| Antibody | Alexa Fluor 488-conjugated Donkey anti-chicken antibody (Donkey polyclonal) | Jackson Immuno Research Labs | Cat# 703-545-155, RRID:AB_2340375 | IF(1:200) |
| Antibody | Alexa Fluor 594-conjugated Donkey anti-mouse antibody (Donkey polyclonal) | Jackson Immuno Research Labs | Cat# 715-585-150, RRID:AB_2340854 | IF(1:200) |
| Antibody | Alexa Fluor 647-conjugated Donkey anti-rabbit antibody (Donkey polyclonal) | Jackson ImmunoResearch Labs | Cat# 711-605-152, RRID:AB_2492288 | IF(1:200) |
| Software, algorithm | IGOR Pro | Wavemetrics | RRID:SCR_000325 | Analysis of calcium imaging and electrophysiology data |
| Software, algorithm | pClamp | Molecular Devices | RRID:SCR_011323 | Signal digitization. software, algorithm |
| | MATLAB | MathWorks | RRID:SCR_001622 | Stimulus generation and presentation |
| Software, algorithm | LabView | National Instruments | RRID:SCR_014325 | Control of microscope and other hardware |
| Software, algorithm | Fiji | http://fiji.sc | RRID:SCR_002285 | Processing and analysis of images from immunohistochemistry |

## Animals

Detailed two-photon imaging experiments were conducted on seven GP4.3 transgenic animals (*Dana et al., 2014*) (three in C57BL/6J background and four in B6CBAF1/J background) and six F1 progeny between Gad2-IRES-cre (*Taniguchi et al., 2011*), and Ai96 reporter line (B6;129S6-*Gt (ROSA)26Sor$^{tm96(CAG-GCaMP6s)Hze}$*/J) (*Madisen et al., 2015*). We will refer to the Gad2-IRES-Cre x Ai96 cross as Gad2;Ai96. Postnatal age at recordings ranged between 11–35 weeks (median: 19; Q1: 15; Q3: 23). Ground-truth juxtacellular recordings were performed on four GP4.3 and three Gad2;Ai96 animals. Immunohistochemistry for cell counting was performed on three GP4.3 and three Gad2;Ai96 animals.

GP4.3 animals were originally obtained from the Jackson Laboratory (C57BL/6J-Tg(Thy1-GCaMP6s)GP4.3Dkim/J; JAX stock #024275); they were maintained in a heterozygous state by back-crossing to C57BL/6J from Charles Rivers (JAX C57BL/6J). To create GP4.3 animals with B6CBAF1/J background, heterozygous GP4.3 in C57BL/6J background were crossed with CBA/JRj mice from Janvier. Gad2-IRES-Cre (originally STOCK *Gad2$^{tm2(cre)Zjh}$*/J, the Jackson Laboratory) was maintained in homozygous state after >10 generations of backcrossing to C57BL/6J, and re-backcrossed to C57BL/6J every 4–5 generations. Ai96 mice were obtained from Jackson Laboratory with already 3 generations of backcross to C57BL/6J (N3), and were subsequently maintained by backcrossing to

C57BL/6J for 5–7 generations. All experiments complied with the ethical guidelines for laboratory animals within our institute and with European guidelines, and were approved by the animal ethical committee of the Erasmus MC.

## Surgery

Animals were anaesthetised through respiratory intake of isoflurane and maintained at surgical level of anaesthesia, assessed through the hind limb withdrawal reflex. A heating pad with rectal feedback probe (40-90-8C; FHC, Bowdoinham, ME, USA) was used to maintain body core temperature at 36–37°C. Eye ointment (Duratears; Alcon Nederland, Gorinchem, The Netherlands) was used to keep the eyes moist during surgery. A bolus of buprenorphine (0.05 mg/kg; Temgesic, Merck Sharp and Dohme, Inc, Kenilworth, NJ, USA) was injected subcutaneously at the beginning of surgery. The skin overlying the IC was incised. Lidocaine (Xylocaine 10%; AstraZeneca, Zoetermeer, The Netherlands) was applied before removing the periosteum and cleaning the skull. After etching the bone surface with phosphoric acid gel (Etch Rite; Pulpdent Corporation, Watertown, MA, USA), a titanium head plate was glued to the cleaned bone above the left IC using dental adhesive (OptiBond FL; Kerr Italia S.r.l., Scafati, SA, Italy) and further secured with dental composite (Charisma; Heraeus Kulzer GmbH, Hanau, Germany).

Through an opening in the head plate, a craniotomy of 3 mm diameter centered at one of the ICs was made by thinning and removing the skull bone. A cranial window, made by gluing a 3 mm cover slip (CS-3R-0; Warner Instrument Inc, Hamden, CT, USA) on a custom built, 500 µm thick steel ring with UV-cured optical adhesive (NOA68; Norland Products), was installed over the exposed brain surface and secured with superglue. Each animal was allowed to recover for at least two days before the first measurements. For *in vivo* electrophysiology, the cranial window construct was gently removed, and the dura mater covering the IC and part of the cerebellum was carefully punctured and removed with a pair of fine forceps.

After all recordings had been done, animals received an intraperitoneal injection of pentobarbital (300 mg/kg) and were perfused transcardially, first with physiological saline (Baxter Healthcare, Zurich, Switzerland), followed by 4% paraformaldehyde (PFA) in 0.1 M phosphate buffer (PB; 4% PFA in PB, pH 7.4; Merck).

## Two-photon imaging

For two-photon imaging of the IC, a 20X water-immersion objective (LUMPlanFl/IR, 20X, NA: 0.95; Olympus Corporation, Tokyo, Japan) on a custom-built two-photon microscope was used, except for simultaneous juxtacellular recordings, for which a long working distance 40x objective (LUMPlanFl/IR, 40X, NA: 0.80, Olympus Corporation, Tokyo, Japan) was used. Excitation light was provided by a MaiTai Ti:Sapphire laser (Spectra Physics Lasers, Mountain View, CA, USA) tuned to a wavelength of 920 nm. A layer of Ringer solution was put between the objective and the cranial window. GCaMP6s fluorescence was captured by a photomultiplier tube (H6780-20, Hamamatsu, Japan) after a barrier filter at 720 nm (FF01-720/SP-25; Semrock), a secondary dichroic at 558 nm and a green bandpass filter centered at 510 nm (bandwidth: 84 nm; FF01-510/84-25; Semrock); AlexaFluor 594 fluorescence was captured at the second channel with a red bandpass filter centered at 630 nm (bandwidth: 60 nm; D630/60; Chroma). Images (256 × 128 pixels) were collected at 9 Hz (2 µs/pixel; 1–2 µm/pixel). To minimize acoustic noise from scanning, a sinusoidal waveform was used for the X Galvo-scanner (for acoustic spectra see *Figure 2—figure supplement 1*). Data were acquired in the middle 80% of the sinusoidal waveform to minimize nonlinearity. Multiple regions were imaged sequentially in multiple sessions in awake, head-fixed animals. For depth estimation, a Z-stack was acquired over a 512 × 512 µm area at 1 × 1 × 1 µm voxel size. Depth of each imaged area was estimated by measuring the Z-distance from the pia surface. Imaged neurons lay between 15–155 µm from the pia surface. The relative position of each region was tracked through the micromanipulator (MP-285, Sutter Instrument) that controlled the microscope objective. The positions of imaged areas were further aligned across animals to a common coordinate using the midline, lateral extreme and anterior-posterior extremes of the IC as anatomical landmarks in bright field images (SZ 61,Olympus) of the cranial window.

Frame timing of the scanner, timing of the sound stimuli, and animal movements were digitized using a Digidata 1440A (Molecular Devices, Sunnyvale, CA, USA) with Clampex v. 10.3 (Molecular Devices, Sunnyvale, CA, USA).

## Juxtacellular recording

*In vivo* juxtacellular recordings were made under two-photon guidance (*Kitamura et al., 2008*). Glass pipettes were pulled from 1.5 mm, thick-walled borosilicate capillaries (Hilgenberg, Malsfeld, Germany) to 1–2 μm tip diameter (P-97; Sutter Instrument, Novato, CA) and filled with internal solution containing (in mM): potassium gluconate 138, KCl 8, $Na_2$-phosphocreatine 10, Mg-ATP 4, $Na_2$-GTP 0.3, EGTA 0.5, HEPES 10, (pH 7.2 with KOH; Merck). The internal solution also contained 40 μM Alexa Fluor 594 hydrazide. Ringer solution containing 1–2% agarose and (in mM): NaCl 135, KCl 5.4, $MgCl_2$ 1, $CaCl_2$ 1.8, HEPES 5, was applied on the brain surface to reduce movement artefacts. A positive pressure of about 300 mbar was maintained before penetration of the pia surface, and reduced to about 30 mbar upon pia entry; pressure was removed upon cell approach. Electrode resistances were constantly monitored and recording started when resistance increased to >25 MΩ. Juxtacellular potentials were acquired with a MultiClamp 700A amplifier (Molecular Devices, Sunnyvale, CA, USA) in current-clamp mode. Signals were low-pass filtered at 10 kHz (four-pole Bessel filter) and digitized at 25 kHz (Digidata 1322A). Data were recorded with pCLAMP 9.2 (Molecular Devices). In some cases a large current injection (1–6 nA) was used to elicit (positive current) or suppress (negative current) spikes of the cell being recorded (nanostimulation; for example *Houweling et al., 2010*), usually for cells that did not show obvious change in spike rate upon pure-tone stimulations. These data were also used for fitting the fluorescence ground-truth model.

## Sound stimulation

Tone stimuli were generated in MATLAB v7.6.0 (The MathWorks, Natick, MA, USA) and played back via a TDT System3 setup (RX6 processor, PA5 attenuator, ED1 electrostatic speaker driver and two EC1 electrostatic speakers; Tucker Davis Technologies, Alachua, FL, USA). Sound stimuli were presented bilaterally in open field. Sound intensities were calibrated using a condenser microphone (ACO pacific Type 7017; ACO Pacific, Inc, Belmont, CA, USA) connected to a calibrated pre-amplifier and placed at the position of the pinnae.

For measurement of FRAs, 1 s tones (including 2.5 ms cosine-squared rise/decay times) with frequencies between 1 and 64 kHz with three steps per octave were presented at intensities between 30 and 80 dB sound pressure level (dB SPL) in steps of 10 dB. The set of stimuli was presented 6–10 times per experiment each in a pseudorandom order at an inter-stimulus interval of 1.5 s. For juxtacellular recordings, due to the limited recording time tones were only presented at one or two intensities (70 dB SPL; or 50 and 70 dB SPL).

## Behavioral measurements

Visual recordings were made using an RS Miniature CCD Camera (RS Components, Corby, UK) at 3 Hz. The camera was aimed at the animal's head from a roughly right lateral perspective, in order to clearly record the animal's right eye and whiskers. Facial movement was detected by calculating the root-mean-square intensity change between successive frames in a rectangular region behind the whisker pad (facial ROI in *Figure 8B*). General movement of the animal was registered using a piezo-electric motion sensor under the front paws of the mouse (pressure sensor in *Figure 8B*), digitized without further amplification using an AD channel of the Digidata 1440A.

## Immunohistochemistry and cell counting

Gelatin-embedded, 40 μm sections of PFA-fixed mouse brains were stained with the following antibodies: mouse anti-Gad67 (cat. no.: MAB5406; Millipore; 1:1000; RRID: AB_2278725); chicken anti-GFP (cat. no.: GFP-1020; Aves; 1:1000; RRID: AB_10000240); rabbit anti-NeuN (cat. no.: ABN78; Millipore; 1:1000; RRID: AB_10807945); mouse anti-parvalbumin (cat. no.: 235; Swant; 1:7000; RRID: AB_10000343); rabbit anti-calretinin (cat. no.: 7699/4; Swant; 1:5000; RRID: AB_2313763); and Alexa-Fluor-, Cy3- or Cy5-conjugated secondary antibodies (Invitrogen or Jackson ImmunoResearch). To ensure a more homogeneous GAD67 fluorescence for the *post-hoc* immunostaining of imaged brains, the brain slices were incubated twice in the primary antibody solution, each for 1 week at 4°

C. The secondary antibody was also applied twice, but overnight at room temperature. Cell counting was performed manually in FIJI using the Cell Counter plug-in on confocal z-stacks. Confocal images were acquired using a Zeiss LSM700 microscope. Overview epifluorescence images were acquired using a Zeiss AxioImagerM2 equipped with a Zeiss Axiocam 503 mono camera.

## General analysis

Data analysis was performed with Igor Pro (WaveMetrics, Inc, Lake Oswego, OR, USA) using custom written procedures. Two-photon images and behavior video were aligned to ClampEx data using stimulus timing. Whisking behavior was assessed by calculating the root-mean-square of the frame to frame intensity difference in an area at the whisker pad.

Movement artefacts in two-photon images were corrected based on the built-in ImageRegistration operation in Igor Pro, which is based on a published algorithm (*Thévenaz et al., 1998*). Neuronal cell bodies were identified visually based on the average image of the motion-corrected image series, and a higher sampling Z-stack of the area (0.5 × 0.5 µm pixels; 1 µm z steps) taken directly after each experiment. Regions-of-interests (ROIs) were drawn around cell bodies. Average fluorescence values for individual ROIs were corrected for background fluorescence, which was defined as the change in average fluorescence in a 2 µm wide contour surrounding the ROI, excluding any pixel directly belonging to another ROI (*Figure 2D*).

## Analysis of Frequency Response Areas and response classification

Individual baseline fluorescence values of an ROI were measured for each trial by averaging the fluorescence in the 1 s preceding each stimulus onset. To mitigate carry over effects from earlier stimuli caused by the slow kinetics of GCaMP6s, the distribution of these baseline values was fitted with a Gaussian distribution. Any trial with a more extreme average baseline fluorescence than $\mu \pm 3\sigma$ was excluded from further analysis.

To analyze the FRA of each ROI, we isolated the stimulus-related response by extracting the *signal autocorrelation* (*Geis et al., 2011*) by calculating the average Pearson correlation coefficient among fluorescence waveforms to the same stimulus; the fluorescence trace within 1 s of the beginning or end of each stimulus was taken as the stimulus related waveform ($F_i(t)$ for stimulus *i*), that is between -1 s and +2 s of the stimulus onset. For a stimulus that was repeated *n* times, we calculated the signal autocorrelation ($\bar{\rho}_{auto}$) by averaging the Pearson's correlation coefficient ($\rho_{i,j}$) for the $\frac{n(n-1)}{2}$ possible pairs of responses:

$$\bar{\rho}_{auto} = \frac{2}{n(n-1)} \sum_{i=1;i}^{n} \rho_{i,j}$$

where $\rho_{i,j}$ was calculated from fluorescence waveforms at trial *i* and *j* with the StatsCorrelation function in Igor Pro using the standard formula:

$$\rho_{i,j} = \frac{\sum \left( \left( F_i(t) - \bar{F}_i \right) \left( F_j(t) - \bar{F}_j \right) \right)}{\sqrt{\sum \left( F_i(t) - \bar{F}_i \right)^2 \sum \left( F_j(t) - \bar{F}_j \right)^2}}$$

Statistical significance of signal correlation was done by a bootstrap method: for each ROI, a distribution of average Pearson correlation coefficient was constructed by drawing 30,000 random samples of *n* fluorescence trace segments within an experimental session, where *n* is again the number of stimulus repetitions. The p-value of each stimulus was calculated as the fraction of *n*-member samples having a greater (for frequency autocorrelation area; *Geis et al., 2011*) signal autocorrelation than that of the stimulus. The ranked p-values were then tested for significance with α = 0.05 and Holm-Bonferroni correction for the number of different stimuli presented (19 frequencies × 6 intensities=114 for FRA measurements). The characteristic frequency (CF) of an ROI was defined as the sound frequency at which the lowest intensity evoked a significant response. If multiple frequencies evoked significant responses at the lowest level, their geometric mean was taken as CF.

The FRA of a cell is generally continuous in frequency-intensity space. Making use of this, we refined the FRA by quantifying the similarity of responses to "adjacent" stimuli by analogously calculating the *signal crosscorrelation* ($\bar{\rho}_{cross}$).

$$\bar{\rho}_{cross} = \frac{1}{nm} \sum_{i=1}^{n} \sum_{j=1}^{m} \rho_{i,j}$$

where n and m are the number of repetitions of the two adjacent stimuli, leading to n×m possible pairs of responses and $\rho_{i,j}$ is again the Pearson correlation coefficient of the two responses. Note that, different from the cell to cell comparison described in *Geis et al. (2011)*, none of the response pairs here were simultaneous. If the maximum $\bar{\rho}_{cross}$ calculated against up to 8 (i.e. including diagonals) neighbors of a particular stimulus was >0.12, the stimulus was included for classification of its response type. For this purpose, the fluorescence response was calculated for four periods: onset (0-500 ms *re* onset), steady (500-1000 ms *re* onset), offset (0-500 ms *re* offset) and off-late (500-1000 ms *re* offset). We used the following scheme (in pseudocode) to classify individual responses within an FRA as excitatory, inhibitory or offset:

| Response class | Criteria (non-exclusive) |
|---|---|
| Excitation (onset and sustained) | $F_{onset} - F_{baseline} > 2 \times$ s.d. baseline; OR<br>$F_{steady} - F_{baseline} > 2 \times$ s.d. baseline |
| Inhibition | $F_{steady} - F_{baseline} < -2 \times$ s.d. baseline; OR<br>$F_{offset} - F_{baseline} < -2 \times$ s.d. baseline |
| Offset | If (excitation)<br>$F_{off\text{-}late} - F_{offset} > 2$ x s.d. baseline<br>elseif (inhibition)<br>$F_{off\text{-}late} - F_{offset} > 2$ x s.d. baseline; AND<br>$F_{off\text{-}late} > F_{baseline}$<br>else<br>$F_{offset} - F_{steady} > 2$ x s.d. baseline; OR<br>$F_{off\text{-}late} - F_{steady} > 2$ x s.d. baseline |
| Unclassified | If none of the above, for example due to low signal-to-noise ratio |

If at least 25% of all significant responses to frequency-intensity combinations belonged to the same class (e.g. inhibition), we named the FRA after that response (e.g. inhibition FRA). Typically an ROI showed a dominant response class in the majority of its significant responses: 20% of cells showed a dominant class in all significant responses; 54% of cells in ≥75% of significant responses; 92% of cells in ≥50% of significant responses. If more than one class reached the threshold of 25% of the significant responses, the FRA of an ROI was classified as mixed (e.g. *Figure 3I,M*).

Finally, to distinguish between onset and sustained responses, we averaged all excitatory responses of an ROI that were not immediately preceded or followed by any significant responses. The kinetics of sound-evoked responses were fitted by a single exponential function with the form $A(1 - \exp(-t/\tau_{onset}))$. If this onset time constant, $\tau_{onset}$, was >1 s, the ROI response type was designated as sustained, else as onset.

## Orientation of CF gradient

To find the most prominent direction of CF gradient, we parametrized the location of each neuron as a projected distance $r$ from the anatomical origin along a line with an angle $\theta$ with the medial-lateral orientation:

$$r = x\cos\theta + y\sin\theta$$

where x and y are coordinates (in micrometers) of cells along the medial-lateral and anterior-posterior axes, respectively. We then model CF as a polynomial function of $r$:

$$a_0 + a_1 r^1 + a_2 r^2 + a_3 r^3 + a_4 r^4 + \ldots = \sum_{i=0}^{n} a_i r^i$$

This whole function was then fitted to the logarithm of the CF values of the recorded neurons using the Levenberg-Marquardt least-squares method, implemented in Igor Pro, that is:

$$\underset{\theta,\, a_{0-4}}{\arg\min} \sum_{x=1}^{n} \left( \log(\mathrm{CF}_x) - \sum_{i=0}^{4} a_i r^i \right)^2$$

A 4th order polynomial was used in *Figure 6* because increasing the number of exponents did not increase the explained variance further. The best fit parameters were $\theta = 0.875$ rad ($\approx 50°$), $a_0 = 6.95$, $a_1 = -0.0135$ dec $\mu m^{-1}$, $a_2 = 1.95 \times 10^{-5}$ dec $\mu m^{-2}$, $a_3 = -1.11 \times 10^{-8}$ dec $\mu m^{-3}$, $a_4 = 2.17 \times 10^{-12}$ dec $\mu m^{-4}$. Cross-validation (70% fitting/30% test) indicated a slight overfitting. We also considered 2D polynomial fits in the form $\sum a_{ij} x^i y^j$, where $0 \le i + j \le n$, and found that a 3rd order ($n = 3$) fit would be an optimal compromise between bias (underfitting) and variance (overfitting). However, we opted for the single dimension polynomial to avoid potential overinterpretation of the complex contour created by the 2D polynomial.

### Ground-truth and modeling of GCaMP6s fluorescence

Traces from juxtacellular recording were first subjected to a digital DC remove filter which subtracts at each point in time the average potential within $\pm 1$ ms to remove DC drift or offset introduced by nanostimulation (*Houweling et al., 2010*). Traces were blanked around the start and end of each current injection (2 ms before and 3 ms after) to remove stimulus artefacts. Spikes were then detected by a simple thresholding procedure, with spike times defined as the peak time of the spikes, which presumably corresponds to the maximum rate of rise of the action potential.

For spike vs fluorescence model, spike times were converted to number of spikes over time ($n(t)$) at the same sampling rate of the imaging (114.4 ms bins). It was then convolved with a simple ramp-decay kernel ($g(t)$) according to the equations:

$$F(t) = F_0 + n(t) * g(t)$$

$$g(t) = \begin{cases} F_{1AP}\left(\frac{t}{t_{\mathrm{rise}}}\right), & t < t\mathrm{rise} \\ F_{1AP}\left(1 - \exp\left(\frac{t - t_{\mathrm{rise}}}{\tau_{\mathrm{decay}}}\right)\right), & t \ge t\mathrm{rise} \end{cases}$$

where $F_{1AP}$ is the amplitude of the fluorescence change per action potential, $t_{\mathrm{rise}}$ is the rise time of the fluorescence and $\tau_{\mathrm{decay}}$ is the decay time constant for the fluorescence. An offset ($F_0$) is added, which represents the minimal fluorescence at zero spike rate.

This 4-parameter model was fitted to the fluorescence trace using a genetic fitting routine in Igor Pro (Gencurvefit XOP, kindly provided by Andrew Nelson, Australian Nuclear Science and Technology Organization).

## Acknowledgements

We thank Kees Donkersloot and Elize Haasdijk for their excellent technical support and Alba Membrilla Esteban for performing part of the immunohistochemistry analysis, Zhenyu Gao for sharing the design of the chronic cranial window, and Travis Babola and Dwight Bergles for helpful discussions on tonotopy. We thank Douglas Kim and the GENIE Project at the Janelia Research Campus for making the GP4.3 mouse line available.

This work was supported by a NeuroBasic PharmaPhenomics grant (Agentschap NL of the Ministry of Health, Welfare and Sports of the Netherlands) and a TOP subsidy (#91218033) from ZonMW to JGGB. ABW was supported by an Individual Fellowship (660157-OPTIMAPIC) through the Marie Skłodowska-Curie actions (European Commission).

## Additional information

### Funding

| Funder | Grant reference number | Author |
| --- | --- | --- |
| Agentschap NL | FES0908 | J Gerard G Borst |

| European Commission | 660157-OPTIMAPIC | Aaron Benson Wong J Gerard G Borst |
| ZonMw | 91218033 | J Gerard G Borst |

The funders had no role in study design, data collection and interpretation, or the decision to submit the work for publication.

## Author contributions

Aaron Benson Wong, Conceptualization, Formal analysis, Funding acquisition, Writing—original draft, Writing—review and editing; J Gerard G Borst, Conceptualization, Software, Supervision, Funding acquisition, Writing—original draft, Writing—review and editing

## Author ORCIDs

Aaron Benson Wong ⓘD https://orcid.org/0000-0003-1650-2710
J Gerard G Borst ⓘD https://orcid.org/0000-0002-6092-1544

## Ethics

Animal experimentation: All experiments in this study were performed in accordance with the ethical guidelines for laboratory animals within our institute and with European guidelines. The study was carried out under the project license (AVD2016789) approved by the Centrale Commissie Dierproeven (CCD) and the animal ethical committee (Instantie voor Dierenwelzijn; IvD) of the Erasmus MC. All recovery surgeries were performed under general isoflurane anesthesia supplemented with lidocaine, carprofen and buprenorphine as peri-operative analgesics. Terminal transcardiac perfusion was performed under pentobarbital anesthesia. Every effort was made to minimize suffering.

## Decision letter and Author response

Decision letter https://doi.org/10.7554/eLife.49091.045
Author response https://doi.org/10.7554/eLife.49091.046

# Additional files

## Supplementary files

• Transparent reporting form  DOI: https://doi.org/10.7554/eLife.49091.044

## Data availability

Source data files have been provided for Figures 1, 4–8, and Figure 3—figure supplement 1.

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
