## [Decision Letter]

Thank you for submitting your article "Tonotopic and multisensory organization of the mouse dorsal inferior colliculus revealed by two-photon imaging" for consideration by *eLife*. Your article has been reviewed by three peer reviewers, including Brice Bathellier as the Reviewing Editor and Reviewer #1, and the evaluation has been overseen by Andrew King as the Senior Editor. The following individuals involved in review of your submission have agreed to reveal their identity: Johannes C Dahmen (Reviewer #2); Daniel Llano (Reviewer #3).

The reviewers have discussed the reviews with one another and the Reviewing Editor has drafted this decision to help you prepare a revised submission.

Summary:

This manuscript is an interesting and detailed investigation into the functional organization of the dorsal inferior colliculus of the mouse based on results from challenging two-photon calcium imaging experiments in unanesthetised animals. Several interesting results are reported, ranging from the lack of spatial organization of simple temporal envelope features, the prominence of inhibitory responses, the tonotopic organization in the dorsal colliculus and the presence of non-auditory activity related to facial movements. Information about the cellular identity of recorded neurons, as well as juxtacellularly recorded electrical signals and histological data, usefully complement this technically solid study.

While the reviewers are convinced that this paper will be a very valuable resource for researchers with an interest in the auditory system, a number of important concerns regarding the reproducibility of the analysis methods, the conclusions on the tonotopic organization of dorsal IC, the description of activity related to facial movements, and the presentation of some of the key results must be addressed before a final decision can be made. The title and Abstract should also be edited to better account for the different results of the study. The title could be broader, removing any reference to tonotopy and multisensory (which is debatable).

Essential revisions:

1) The authors should make sure to provide reproducible analyses based on a well-described statistical assessment. This is missing in several places:

- It is not clear in the manuscript what criteria are used to define each functional cell type (onset, offset, sustained) and how these criteria are measured and statistically assessed. This is particularly important as several claims of the paper are dependent on these classes. The authors should make sure that this is done in a reproducible manner and provide enough information about the algorithm used for cell classification.

- Figure 4D; "There were more sustained responses in cells with a larger size." This is unclear from the figure. The graph is complex and there is no statistical assessment. Why not just compute the mean size for each class and test for differences?

- Regarding the idea of cell classes, the data in Figure 4 are intriguing, but the figure is still confusing. The Legend shown in Figure 7 should actually appear in Figure 4. Beyond that, it would be helpful to have diagrams, similar to the 4 idealized waveforms shown in Figure 4, to describe the rest of the response types.

2) The analysis of the two horizontal tonotopic gradients in the dorsal part of the IC is interesting but incomplete. The authors state that this analysis is biased by more dense sampling at the center of the IC with deeper recordings here without resolving this bias. We know that the central nucleus of the IC is organised such that neurons with a preference for low frequencies are located at its dorsal tip. In the present manuscript the authors find that in a region that approaches the medial edge of the IC's surface low frequency neurons dominate. This overrepresentation of low frequency neurons seems to become particularly strong as soon as one images just 100um below the brain surface (Figure 6B), where almost no high frequency neurons can be encountered anymore. The most parsimonious explanation for this clustering of low frequency neurons just below the surface of the IC is that these neurons form the most dorsal tip of the central nucleus of the IC as hypothesized by Barnstedt et al., 2015 (who actually also see the continuation of the vertical gradient in the deepest recordings). The authors acknowledge this explanation (e.g. subsection “Spatial distribution of frequency tuning”, third paragraph) but nevertheless choose to regard the low frequency cluster as belonging to the shell of the IC (dorsal/lateral cortex) and go on to interpret the low frequency region as the site of a frequency reversal in a high-low-high frequency map. The authors are correct in saying that a gradient reversal along the dorsal surface of the IC was also observed in one mouse imaged by Barnstedt et al. However, that reversal was actually opposite in sign, i.e. low-high-low rather than the high-low-high reversal reported here.

More analysis is required to substantiate (or not) the discrepancy with Barnstedt's conclusions. As a minimum, the tonotopic gradient should be described in the first 50µm alone (without taking recordings deeper than 50µm into account). But it would be better to show the progressive changes in BF organization as one progresses more deeply into the IC. Does the tonotopy seen at the surface continue to project downward, as it does in the auditory cortex?

Another important analysis is to test if there is any evidence of frequency gradients along the dorsal surface when the low frequency clusters are removed from the analysis (i.e. remove all neurons with a CF of, for example, 8kHz and below). Also, it might be informative to plot some frequency maps from individual mice rather than only aggregated ones.

The authors also state that they have oversampled the center of IC at larger depths. This is important information about the limitations of their dataset. It should, however, not be used as an argument in favor of their hypothesis, as it is done currently.

The dimension of depth is glossed over throughout the manuscript. It should also be addressed more directly for temporal features and non-auditory responses.

3) The analysis of calcium kinetics is not convincing, in particular, the rise time values seem much too large and spread out. This is likely because the equation used is underdetermined (i.e. too many free parameters – 4 instead of 3). The rising phase equation contains a parameter A for amplitude and t_rise_ which are totally redundant (there is an infinity of combination A and t_rise_ that lead to the same curve).

It is necessary to redo this analysis with a 3 parameter kernel, such as F_1AP_*f(t)/f_max_ with f(t)=(exp(-t/t_decay_)-exp(-t/t_rise_)) and f_max_=max(f(t)).

It is also necessary to see how well the model performs by comparing real vs. model traces. This is described in the subsection “Electrophysiological correlates of response classes”, but the fits are not shown.

4) Contrary to what the title suggests, the manuscript tells us little about the relationship between neural activity in the dorsal IC and multisensory input. The authors show that a small proportion of neurons show activity patterns that vary with facial movements. That does not mean that these activity patterns are the consequence of stimulation of somatosensory receptors that results from these movements. However, that seems to be how the authors interpret this result. Provided the ROIs were on the whisker pad (which isn't actually clear from the figures – the ROI shown in Figure 8B appears to have been placed behind the whisker pad rather than on it), the data indicate that the activity of some IC neurons varies with self-generated movements of the whiskers. We do not know, however, whether the changes in neural activity reflect somatosensory stimulation arising from the whisker movement (such as when the whiskers touch a part of the experimental apparatus during those movements) or are related to, for instance, the preparation of the movement, its execution or the product of some change in internal state that accompanies the movement. Therefore, claims that these data indicate sensitivity of IC neurons to somatosensory (or even multisensory) input should be toned down. Unless direct stimulation data are provided, the term 'multisensory' should be removed from the title and the description of the results focused on non-auditory inputs, related to facial movements.

5) It is not clear what the analysis of PV/CR/GCaMP cells contributes to this manuscript. It should be more clearly linked to the main themes of the paper as the other conclusions are reframed ('multisensory', claims on tonotopy). Also the observation that there is strong inhibition should be better emphasized as it is one of the important observations of the paper.

6) Regarding the "modules" seen in Figure 9A, these look very different from the more traditional modules in Figure 9—figure supplement 1, which are larger and stain more strongly for GAD. It is an important distinction, because the presence of modules in the dorsal cortex would be new, as far as we are aware, but these modules look very different than those of Chernock et al. This distinction should be made in the text.

[Editors' note: further revisions were requested prior to acceptance, as described below.]

Thank you for resubmitting your work entitled "Tonotopic and non-auditory organization of the mouse dorsal inferior colliculus revealed by two-photon imaging" for further consideration at *eLife*. Your revised article has been favorably evaluated by Andrew King as the Senior Editor, and a Reviewing Editor.

The manuscript has been improved but there are some remaining issues that need to be addressed before acceptance, as outlined below:

1) Figure 6—figure supplement 9, very convincingly addresses the question whether the tonotopy observed in this study is depth-dependent within the range of depths that were extensively sampled. Looking at it, it becomes evident that depth is not an issue. This figure should be presented as a new panel of the main Figure 6 not as a figure supplement.

Also, the caption of this figure should be changed. The current title does not describe it appropriately. Something like "tonotopy across different recording depths" would be better suited.

It is also necessary to mitigate the statement about horizontal localisation biases (more central) for high depth. It seems that just very few recordings were done below 100µm.

2) It seems that the acronym CF is nowhere defined. Please do so.

It is also important to describe in the method (Analysis) how the CF was computed, particularly as previous calcium imaging studies have focused on the best frequency (i.e., the sound frequency at which the strongest response was elicited).

3) Please check again for typos and duplications in the revised text. Here a few examples:

"For somatosensory inputs the dominant effect of somatosensory inputs appears to be inhibitory and only a minority of cells have been shown to respond to unimodal tactile stimuli" > Remove one 'somatosensory inputs'

"appreciated in neurons in individual animals" > remove 'in neurons'

"responses to "adjacent" stimuli by analogously calculated" > calculating.

---

## [Author Response]

Essential revisions:1) The authors should make sure to provide reproducible analyses based on a well-described statistical assessment. This is missing in several places:- It is not clear in the manuscript what criteria are used to define each functional cell type (onset, offset, sustained) and how these criteria are measured and statistically assessed. This is particularly important as several claims of the paper are dependent on these classes. The authors should make sure that this is done in a reproducible manner and provide enough information about the algorithm used for cell classification.

We thank the reviewers for this helpful comment. We have formalized the classification of our cells and in the Materials and methods section of our revised manuscript we have included the objective criteria for classifying the response of a cell. Briefly, the similarity of responses to the same stimulus determines whether a sound-evoked response is present. The average fluorescence change at different time periods determines the response class to a particular stimulus. Most ROIs showed a dominant response class (excitation, inhibition or offset) in the majority of its significant responses in an FRA (Author response image 1). If more than one class reached a threshold of 25% of the significant responses, the FRA of an ROI was classified as mixed. Finally, the distinction between onset and sustained excitation is based on the time constant of a single exponential fit to the fluorescence during the stimulus period.

**Author response image 1. respfig1:** Histogram of the fraction of significant responses showing the largest response class (dominant class) in the FRA of all responsive ROIs.

We have reclassified all ROIs based on these criteria and updated the figures accordingly. The proportion of different response classes remained similar to the previous version, with slightly more cells processing mixed response types (excitation + inhibition; inhibition + offset, etc.). We are confident that the inclusion of these objective criteria will aid in the reproducibility of our results, and will be useful for future studies.

- Figure 4D; "There were more sustained responses in cells with a larger size." This is unclear from the figure. The graph is complex and there is no statistical assessment. Why not just compute the mean size for each class and test for differences?

We again thank the reviewers for the helpful suggestion. The cell sizes are now plotted in Figure 4C both as a function of response class as well as neurochemical identity (genotype of animal). Additional analysis made clear that the apparent enrichment of sustained responses in bigger cells is a consequence of GABAergic cells being bigger and at the same time a larger fraction of GABAergic cells being classified as sustained. We now include statistical tests for significance.

- Regarding the idea of cell classes, the data in Figure 4 are intriguing, but the figure is still confusing. The Legend shown in Figure 7 should actually appear in Figure 4. Beyond that, it would be helpful to have diagrams, similar to the 4 idealized waveforms shown in Figure 4, to describe the rest of the response types.

We have adopted the suggestion to plot cell size as function of response class as well as neurochemical identity in Figure 4. Idealized waveforms are added for mixed classes to the legend in Figure 7.

2) The analysis of the two horizontal tonotopic gradients in the dorsal part of the IC is interesting but incomplete. The authors state that this analysis is biased by more dense sampling at the center of the IC with deeper recordings here without resolving this bias. We know that the central nucleus of the IC is organised such that neurons with a preference for low frequencies are located at its dorsal tip. In the present manuscript the authors find that in a region that approaches the medial edge of the IC's surface low frequency neurons dominate. This overrepresentation of low frequency neurons seems to become particularly strong as soon as one images just 100um below the brain surface (Figure 6B), where almost no high frequency neurons can be encountered anymore. The most parsimonious explanation for this clustering of low frequency neurons just below the surface of the IC is that these neurons form the most dorsal tip of the central nucleus of the IC as hypothesized by Barnstedt et al., 2015 (who actually also see the continuation of the vertical gradient in the deepest recordings). The authors acknowledge this explanation (e.g. subsection “Spatial distribution of frequency tuning”, third paragraph) but nevertheless choose to regard the low frequency cluster as belonging to the shell of the IC (dorsal/lateral cortex) and go on to interpret the low frequency region as the site of a frequency reversal in a high-low-high frequency map. The authors are correct in saying that a gradient reversal along the dorsal surface of the IC was also observed in one mouse imaged by Barnstedt et al. However, that reversal was actually opposite in sign, i.e. low-high-low rather than the high-low-high reversal reported here.More analysis is required to substantiate (or not) the discrepancy with Barnstedt's conclusions. As a minimum, the tonotopic gradient should be described in the first 50µm alone (without taking recordings deeper than 50µm into account). But it would be better to show the progressive changes in BF organization as one progresses more deeply into the IC. Does the tonotopy seen at the surface continue to project downward, as it does in the auditory cortex?

We thank the reviewers for the suggestion. We have performed a fit on cells that lie within 50 µm of the dorsal surface and have included this as a supplement to Figure 6 (Figure 6—figure supplement 7). This restriction did not qualitatively change the results of the fit, showing a central region of lower frequencies and higher frequencies towards medial and lateral extremes.

To illustrate the bias for deeper cells in the low frequency region, we supply another supplementary figure (Figure 6—figure supplement 8), in which we plotted the depth and projected horizontal distance *r* of each cell, color-coded with their CF.

Another important analysis is to test if there is any evidence of frequency gradients along the dorsal surface when the low frequency clusters are removed from the analysis (i.e. remove all neurons with a CF of, for example, 8kHz and below).

We repeated a gradient fit without the central low frequency cluster (indicated in Author response image 2 300 μm band around the frequency minimum, r = 525 – 825 μm), and fitted the medial and lateral regions, separately. In both the medial (Author response image 3) and lateral (Author response image 4) portions, a gradient of characteristic frequencies matching that of the overall fit was obtained. Excluding all cells with CF < 8 kHz, as suggested by one of the reviewers, removed more than half (52%) of the cells from analysis (see Author response image 2). Nevertheless, we did perform a fit on the remaining cells with CF >= 8 kHz (Author response image 5), which resulted in a plateau of about 13 kHz in the central region, showing the importance of cells with a low CF for the tonotopical gradient.

**Author response image 2. respfig2:** Defining medial and lateral subregions for analysis.

**Author response image 3. respfig3:** Polynomial fit to characteristic frequencies of medial cells.

**Author response image 4. respfig4:** Polynomial fit to characteristic frequencies of lateral cells.

**Author response image 5. respfig5:** Polynomial fit to characteristic frequencies (CF) of cells with >= 8 kHz CF.

Another important analysis is to test if there is any evidence of frequency gradients along the dorsal surface when the low frequency clusters are removed from the analysis (i.e. remove all neurons with a CF of, for example, 8kHz and below). Also, it might be informative to plot some frequency maps from individual mice rather than only aggregated ones.

We now included the CF distribution of 6 mice as supplements to figure 6 (Figure 6—figure supplements 1-6).

The authors also state that they have oversampled the center of IC at larger depths. This is important information about the limitations of their dataset. It should, however, not be used as an argument in favor of their hypothesis, as it is done currently.The dimension of depth is glossed over throughout the manuscript. It should also be addressed more directly for temporal features and non-auditory responses.

In our submitted version we wrote “There was, however, an over-representation of low CF neurons among the deepest imaged (121 – 160 μm deep, Figure 6B). We attribute this to a sampling bias where the low frequency region coincided with the center of the cranial window, thus providing the best optical access for deeper imaging.” This was intended to reflect what we see as a limitation of the data, i.e. that we were unable to image deep at the edges of the cranial window (see Figure 6—figure supplement 9). We rephrased it to emphasize this point. This limitation also precluded a detailed analysis of the effects of depth *per se* as we would be unable to disentangle depth and region effects.

These sentences were now included in the manuscript: “We did observe a small but significant correlation between depth and log(CF) (*r* = -0.22; n=799; p=3×10^-10^).” “Restricting the analysis to cells in the central strip (525-725 μm) reduced the correlation coefficient, but it remained significant (*r* = -0.17, n = 295; p = 0.003). (Figure 6—figure supplement 7)”.

3) The analysis of calcium kinetics is not convincing, in particular, the rise time values seem much too large and spread out. This is likely because the equation used is underdetermined (i.e. too many free parameters – 4 instead of 3). The rising phase equation contains a parameter A for amplitude and t_rise_ which are totally redundant (there is an infinity of combination A and t_rise_ that lead to the same curve).It is necessary to redo this analysis with a 3 parameter kernel, such as F_1AP_*f(t)/f_max_ with f(t)=(exp(-t/t_decay_)-exp(-t/t_rise_)) and f_max_=max(f(t)).It is also necessary to see how well the model performs by comparing real vs. model traces. This is described in the subsection “Electrophysiological correlates of response classes” but the fits are not shown.

We thank the reviewers for spotting this typo. In an earlier version of the manuscript the amplitude of the kernel was labelled as “A”, which was later renamed to the more descriptive F_1AP_. The “A” and “F_1AP_” were thus exactly the same parameter, and we had indeed performed the analysis with a 3-parameter kernel. In fact, we had performed a series of fits using models of different levels of complexity and came to the same conclusion as the reviewer that models more complex than the 3-parameter kernel model result in overfitting of the data. We therefore chose to use the 3-parameter kernel model (4 parameters in total).

The fit parameters of individual data points, and the variance explained by the model are now provided as a source data file (Figure 5—source data 1).

4) Contrary to what the title suggests, the manuscript tells us little about the relationship between neural activity in the dorsal IC and multisensory input. The authors show that a small proportion of neurons show activity patterns that vary with facial movements. That does not mean that these activity patterns are the consequence of stimulation of somatosensory receptors that results from these movements. However, that seems to be how the authors interpret this result. Provided the ROIs were on the whisker pad (which isn't actually clear from the figures – the ROI shown in Figure 8B appears to have been placed behind the whisker pad rather than on it), the data indicate that the activity of some IC neurons varies with self-generated movements of the whiskers. We do not know, however, whether the changes in neural activity reflect somatosensory stimulation arising from the whisker movement (such as when the whiskers touch a part of the experimental apparatus during those movements) or are related to, for instance, the preparation of the movement, its execution or the product of some change in internal state that accompanies the movement. Therefore, claims that these data indicate sensitivity of IC neurons to somatosensory (or even multisensory) input should be toned down. Unless direct stimulation data are provided, the term 'multisensory' should be removed from the title and the description of the results focused on non-auditory inputs, related to facial movements.

We agree with this point and we have rewritten the paper including the title to reflect this.

The ROI shown in Figure 8B is indeed behind the whisker pad, but it is a good position to detect whisker movements. Similar results would have been obtained with an ROI centered on the whisker pad itself, since the movements in these two regions were highly correlated (Author response image 6).

**Author response image 6. respfig6:** Highly correlated movement in facial and whisker pad areas. Left, two different analysis regions-of-interest (ROIs). Right: Changes in intensity at the facial ROI (Y-axis, as analyzed in the manuscript) is highly correlated with that in the whisker pad region (X-axis). Blue: regression line.

5) It is not clear what the analysis of PV/CR/GCaMP cells contributes to this manuscript. It should be more clearly linked to the main themes of the paper as the other conclusions are reframed ('multisensory', claims on tonotopy).

We believe that our conclusions depend on which cells in both mouse lines are expressing the reporter. We therefore view the characterization of these cells as an important step to allow the interpretation of our Ca^2+^ imaging results. We have stressed this more in the revised version of our manuscript.

Also the observation that there is strong inhibition should be better emphasized as it is one of the important observations of the paper.

This is emphasized more in the revised version of our manuscript.

6) Regarding the "modules" seen in Figure 9A, these look very different from the more traditional modules in Figure 9—figure supplement 1, which are larger and stain more strongly for GAD. It is an important distinction, because the presence of modules in the dorsal cortex would be new, as far as we are aware, but these modules look very different than those of Chernock et al. This distinction should be made in the text.

Indeed, the fainter staining in the slices shown in Figure 9A and 9E had initially led us to wonder whether they belonged to the same modular structures as presented by Chernock et al., 2004, and Lesicko et al., 2016. We looked carefully at the serial sections presented in Figure 9—figure supplement 1 (all sections were consecutive). In the more ventral slices, stronger GAD67-stained modules exist, which have weaker calretinin staining (Figure 9—figure supplement 2), and looked similar to those modules that have been presented in the literature. When we followed these modules from ventral to dorsal slices, we found that they were contiguous with the fainter GAD staining pointed out by the arrowheads in Figure 9A and 9E. We attribute the lower GAD67 intensity under epifluorescence to the fact that in the most dorsal/superficial slices the modules were cut either tangentially or at an oblique angle, rather than transversely, as in the more ventral slices, and thus do not occupy the whole thickness of the 40 μm slice. Chernock et al., 2004, and Lesicko et al., 2016, may have missed the rostromedial modules because of their coronal sectioning plane, in which the modules would have been cut tangentially.

We now updated the Figure 9—figure supplement 1 with numbered arrowheads in consecutive slices to help readers follow the GAD67-dense modules along the dorsal-ventral direction. We now also provide additional supplements (Figure 9—figure supplements 4-6) with consecutive slices correspond to the slice shown in Figure 9E.

[Editors' note: further revisions were requested prior to acceptance, as described below.]The manuscript has been improved but there are some remaining issues that need to be addressed before acceptance, as outlined below:1) Figure 6—figure supplement 9, very convincingly addresses the question whether the tonotopy observed in this study is depth-dependent within the range of depths that were extensively sampled. Looking at it, it becomes evident that depth is not an issue. This figure should be presented as a new panel of the main Figure 6 not as a figure supplement.

This supplementary figure is now Figure 6E.

Also, the caption of this figure should be changed. The current title does not describe it appropriately. Something like "tonotopy across different recording depths" would be better suited.

We changed the caption to “Depth dependence of CFs along the 50° line”.

It is also necessary to mitigate the statement about horizontal localisation biases (more central) for high depth. It seems that just very few recordings were done below 100µm.

We revised this section, and we now stress that most recordings were <100 µm from pia surface.

2) It seems that the acronym CF is nowhere defined. Please do so.

Corrected.

It is also important to describe in the method (Analysis) how the CF was computed, particularly as previous calcium imaging studies have focused on the best frequency (i.e., the sound frequency at which the strongest response was elicited).

We added: “The characteristic frequency (CF) of an ROI was defined as the sound frequency at which the lowest intensity evoked a significant response. If multiple frequencies evoked significant responses at the lowest level, their geometric mean was taken as CF” to the Materials and methods section.

3) Please check again for typos and duplications in the revised text.

We checked again and made some small, textual, tracked changes in the revised manuscript.

Here a few examples:"For somatosensory inputs the dominant effect of somatosensory inputs appears to be inhibitory and only a minority of cells have been shown to respond to unimodal tactile stimuli" > Remove one 'somatosensory inputs'

Corrected.

"appreciated in neurons in individual animals" > remove 'in neurons'

Corrected.

"responses to "adjacent" stimuli by analogously calculated" > calculating.

Corrected.